# Optimal Sparse Decision Trees

**Xiyang Hu[1], Cynthia Rudin[2], Margo Seltzer[3]** [*]
[1]Carnegie Mellon University, `xiyanghu@cmu.edu`
[2]Duke University, `cynthia@cs.duke.edu`
[3]The University of British Columbia, `mseltzer@cs.ubc.ca`

## Abstract

Decision tree algorithms have been among the most popular algorithms for interpretable (transparent) machine learning since the early 1980's. The problem that has plagued decision tree algorithms since their inception is their lack of optimality, or lack of guarantees of closeness to optimality: decision tree algorithms are often greedy or myopic, and sometimes produce unquestionably suboptimal models. Hardness of decision tree optimization is both a theoretical and practical obstacle, and even careful mathematical programming approaches have not been able to solve these problems efficiently. This work introduces the first practical algorithm for optimal decision trees for binary variables. The algorithm is a co-design of analytical bounds that reduce the search space and modern systems techniques, including data structures and a custom bit-vector library. Our experiments highlight advantages in scalability, speed, and proof of optimality.

## 1 Introduction

Interpretable machine learning has been growing in importance as society has begun to realize the dangers of using black box models for high stakes decisions: complications with confounding have haunted our medical machine learning models [22], bad predictions from black boxes have announced to millions of people that their dangerous levels of air pollution were safe [15], high-stakes credit risk decisions are being made without proper justification, and black box risk predictions have been wreaking havoc with the perception of fairness of our criminal justice system [10]. In all of these applications – medical imaging, pollution modeling, recidivism risk, credit scoring – accurate interpretable models have been created (by the Center for Disease Control and Prevention, Arnold Foundation, and others). However, such interpretable-yet-accurate models are not generally easy to construct. If we want people to replace their black box models with interpretable models, the tools to build these interpretable models must first exist.

Decision trees are one of the leading forms of interpretable models. Despite several attempts over the last several decades to improve the optimality of decision tree algorithms, the CART [7] and C4.5 [19] decision tree algorithms (and other greedy tree-growing variants) have remained as dominant methods in practice. CART and C4.5 grow decision trees from the top down without backtracking, which means that if a suboptimal split was introduced near the top of the tree, the algorithm could spend many extra splits trying to undo the mistake it made at the top, leading to less-accurate and less-interpretable trees. Problems with greedy splitting and pruning have been known since the early 1990's, when mathematical programming tools had started to be used for creating optimal binary-split decision trees [3, 4], in a line of work [5, 6, 16, 18] until the present [20]. However, these techniques use all-purpose optimization toolboxes and tend not to scale to realistically-sized problems unless simplified to trees of a specific form. Other works [11] make overly strong assumptions (e.g., independence of all variables) to ensure optimal trees are produced using greedy algorithms.

---

[*]Authors are listed alphabetically.

We produce optimal sparse decision trees taking a different approach than mathematical programming, greedy methods, or brute force. We find optimal trees according to a regularized loss function that balances accuracy and the number of leaves. Our algorithm is computationally efficient due to a collection of analytical bounds to perform massive pruning of the search space. Our implementation uses specialized data structures to store intermediate computations and symmetries, a bit-vector library to evaluate decision trees more quickly, fast search policies, and computational reuse. Despite the hardness of finding optimal solutions, our algorithm is able to locate optimal trees and prove optimality (or closeness of optimality) in reasonable amounts of time for datasets of the sizes used in the criminal justice system (tens of thousands or millions of observations, tens of features).

Because we find provably optimal trees, our experiments show where previous studies have claimed to produce optimal models yet failed; we show specific cases where this happens. We test our method on benchmark data sets, as well as criminal recidivism and credit risk data sets; these are two of the high-stakes decision problems where interpretability is needed most in AI systems. We provide ablation experiments to show which of our techniques is most influential at reducing computation for various datasets. As a result of this analysis, we are able to pinpoint possible future paths to improvement for scalability and computational speed. Our contributions are: (1) The first practical optimal binary-variable decision tree algorithm to achieve solutions for nontrivial problems. (2) A series of analytical bounds to reduce the search space. (3) Algorithmic use of a tree representation using only its leaves. (4) Implementation speedups saving 97% run time. (5) We present the first optimal sparse binary split trees ever published for the COMPAS and FICO datasets.

The code and the supplementary materials are available at `https://github.com/xiyanghu/OSDT`.

## 2   Related Work

Optimal decision trees have a quite a long history [3], so we focus on closely related techniques. There are efficient algorithms that claim to generate optimal sparse trees, but do not optimally balance the criteria of optimality and sparsity; instead they pre-specify the topology of the tree (*i.e.*, they know *a priori* exactly what the structure of the splits and leaves are, even though they do not know which variables are split) and only find the optimal tree of the given topology [16]. This is not the problem we address, as we do not know the topology of the optimal tree in advance. The most successful algorithm of this variety is BinOCT [20], which searches for a complete binary tree of a given depth; we discuss BinOCT shortly. Some exploration of learning optimal decision trees is based on boolean satisfiability (SAT) [17], but again, this work looks only for the optimal tree of a given number of nodes. The DL8 algorithm [18] optimizes a ranking function to find a decision tree under constraints of size, depth, accuracy and leaves. DL8 creates trees from the bottom up, meaning that trees are assembled out of all possible leaves, which are itemsets pre-mined from the data [similarly to 2]. DL8 does not have publicly available source code, and its authors warn about running out of memory when storing all partially-constructed trees. Some works consider oblique trees [6], where splits involve several variables; oblique trees are not addressed here, as they can be less interpretable.

The most recent mathematical programming algorithms are OCT [5] and BinOCT [20]. Example figures from the OCT paper [5] show decision trees that are clearly suboptimal. However, as the code was not made public, the work in the OCT paper [5] is not easily reproducible, so it is not clear where the problem occurred. We discuss this in Section §4. Verwer and Zhang's mathematical programming formulation for BinOCT is much faster [20], and their experiments indicate that BinOCT outperforms OCT, but since BinOCT is constrained to create complete binary trees of a given depth rather than optimally sparse trees, it sometimes creates unnecessary leaves in order to complete a tree at a given depth, as we show in Section §4. BinOCT solves a dramatically easier problem than the method introduced in this work. As it turns out, the search space of perfect binary trees of a given depth is much smaller than that of binary trees with the same number of leaves. For instance, the number of different unlabeled binary trees with 8 leaves is $Catalan(7) = 429$, but the number of unlabeled perfect binary trees with 8 leaves is only 1. In our setting, we penalize (but do not fix) the number of leaves, which means that our search space contains all trees, though we can bound the maximum number of leaves based on the size of the regularization parameter. Therefore, our search space is much larger than that of BinOCT.

Our work builds upon the CORELS algorithm  [1, 2, 13] and its predecessors [14, 21], which create optimal decision lists (rule lists). Applying those ideas to decision trees is nontrivial. The rule list

Search Space of CORELS and Decision Trees

| $d$ | $p = 10$ | | $p = 20$ | |
|---|---|---|---|---|
| | Rule Lists | Trees | Rule Lists | Trees |
| 1 | $5.500 \times 10^1$ | $1.000 \times 10^1$ | $2.100 \times 10^2$ | $2.000 \times 10^1$ |
| 2 | $3.025 \times 10^3$ | $1.000 \times 10^3$ | $4.410 \times 10^4$ | $8.000 \times 10^3$ |
| 3 | $1.604 \times 10^5$ | $5.329 \times 10^6$ | $9.173 \times 10^6$ | $9.411 \times 10^8$ |
| 4 | $8.345 \times 10^6$ | $9.338 \times 10^{20}$ | $1.898 \times 10^9$ | $9.204 \times 10^{28}$ |
| 5 | $4.257 \times 10^8$ | "Inf" | $3.911 \times 10^{11}$ | "Inf" |

Table 1: Search spaces of rule lists and decision trees with number of variables $p = 10, 20$ and depth $d = 1, 2, 3, 4, 5$. The search space of the trees explodes in comparison.

optimization problem is much easier, since the rules are pre-mined (there is no mining of rules in our decision tree optimization). Rule list optimization involves selecting an optimal subset and an optimal permutation of the rules in each subset. Decision tree optimization involves considering every possible split of every possible variable and every possible shape and size of tree. This is an exponentially harder optimization problem with a huge number of symmetries to consider. In addition, in CORELS, the maximum number of clauses per rule is set to be $c = 2$. For a data set with $p$ binary features, there would be $D = p + \binom{p}{2}$ rules in total, and the number of distinct rule lists with $d_r$ rules is $P(D, d_r)$, where $P(m, k)$ is the number of $k$-permutations of $m$. Therefore, the search space of CORELS is $\sum_{d_r=1}^{d-1} P(D, d_r)$. But, for a full binary tree with depth $d_t$ and data with $p$ binary features, the number of distinct trees is:

$$N_{d_t} = \sum_{n_0=1}^{1} \sum_{n_1=1}^{2^{n_0}} \cdots \sum_{n_{d_t-1}=1}^{2^{n_{d_t}-2}} p \times \binom{2^{n_0}}{n_1} (p-1)^{n_1} \times \ldots \times \binom{2^{n_{d_t}-2}}{n_{d_t-1}} (p-(d_t-1))^{n_{d_t}-1}, \quad (1)$$

and the search space of decision trees up to depth $d$ is $\sum_{d_t=1}^{d} N_{d_t}$. Table 1 shows how the search spaces of rule lists and decision trees grow as the tree depth increases. The search space of the trees is massive compared to that of the rule lists.

Applying techniques from rule lists to decision trees necessitated new designs for the data structures, splitting mechanisms and bounds. An important difference between rule lists and trees is that during the growth of rule lists, we add only one new rule to the list each time, but for the growth of trees, we need to split existing leaves and add a new *pair* of leaves for each. This leads to several bounds that are quite different from those in CORELS, *i.e.*, Theorem 3.4, Theorem 3.5 and Corollary E.1, which consider a pair of leaves rather than a single leaf. In this paper, we introduce bounds only for the case of one split at a time; however, in our implementation, we can split more than one leaf at a time, and the bounds are adapted accordingly.

## 3  Optimal Sparse Decision Trees (OSDT)

We focus on binary classification, although it is possible to generalize this framework to multiclass settings. We denote training data as $\{(x_n, y_n)\}_{n=1}^{N}$, where $x_n \in \{0, 1\}^M$ are binary features and $y_n \in \{0, 1\}$ are labels. Let $\mathbf{x} = \{x_n\}_{n=1}^{N}$ and $\mathbf{y} = \{y_n\}_{n=1}^{N}$, and let $x_{n,m}$ denote the $m$-th feature of $x_n$. For a decision tree, its leaves are conjunctions of predicates. Their order does not matter in evaluating the accuracy of the tree, and a tree grows only at its leaves. Thus, within our algorithm, we represent a tree as a collection of leaves. A leaf set $d = (p_1, p_2, \ldots, p_H)$ of length $H \geq 0$ is an $H$-tuple containing $H$ distinct leaves, where $p_k$ is the classification rule of the path from the root to leaf $k$. Here, $p_k$ is a Boolean assertion, which evaluates to either true or false for each datum $x_n$ indicating whether it is classified by leaf $k$. Here, $\hat{y}_k^{(\text{leaf})}$ is the label for all points so classified.

We explore the search space by considering which leaves of the tree can be beneficially split. The leaf set $d = (p_1, p_2, \ldots, p_K, p_{K+1}, \ldots, p_H)$ is the $H$-leaf tree, where the first $K$ leaves may not to be split, and the remaining $H - K$ leaves can be split. We alternately represent this leaf set as $d = (d_{un}, \delta_{un}, d_{\text{split}}, \delta_{\text{split}}, K, H)$, where $d_{un} = (p_1, \ldots, p_K)$ are the unchanged leaves of $d$, $\delta_{un} = (\hat{y}_1^{(\text{leaf})}, \ldots, \hat{y}_K^{(\text{leaf})}) \in \{0, 1\}^K$ are the predicted labels of leaves $d_{un}$, $d_{\text{split}} = (p_{K+1}, \ldots, p_H)$ are the leaves we are going to split, and $\delta_{\text{split}} = (\hat{y}_{K+1}^{(\text{leaf})}, \ldots, \hat{y}_H^{(\text{leaf})}) \in \{0, 1\}^{H-K}$ are the predicted labels of leaves $d_{\text{split}}$. We call $d_{un}$ a $K$-prefix of $d$, which means its leaves are a size-$K$ unchanged subset of $(p_1, \ldots, p_K, \ldots, p_H)$. If we have a new prefix $d'_{un}$, which is a superset of $d_{un}$, *i.e.*, $d'_{un} \supseteq d_{un}$, then

we say $d'_{un}$ starts with $d_{un}$. We define $\sigma(d)$ to be all descendents of $d$:

$$\sigma(d) = \{(d'_{un}, \delta'_{un}, d'_{\text{split}}, \delta'_{\text{split}}, K', H_{d'}) : d'_{un} \supseteq d_{un}, d' \supset d\}. \tag{2}$$

If we have two trees $d = (d_{un}, \delta_{un}, d_{\text{split}}, \delta_{\text{split}}, K, H)$ and $d' = (d'_{un}, \delta'_{un}, d'_{\text{split}}, \delta'_{\text{split}}, K', H')$, where $H' = H + 1, d' \supset d, d'_{un} \supseteq d_{un}$, *i.e.*, $d'$ contains one more leaf than $d$ and $d'_{un}$ starts with $d_{un}$, then we define $d'$ to be a child of $d$ and $d$ to be a parent of $d'$.

Note that two trees with identical leaf sets, but different assignments to $d_{un}$ and $d_{\text{split}}$, are *different* trees. Further, a child tree can *only* be generated through splitting leaves of its parent tree within $d_{\text{split}}$.

A tree $d$ classifies datum $x_n$ by providing the label prediction $\hat{y}_k^{(\text{leaf})}$ of the leaf whose $p_k$ is true for $x_n$. Here, the leaf label $\hat{y}_k^{(\text{leaf})}$ is the majority label of data captured by the leaf $k$. If $p_k$ evaluates to true for $x_n$, we say the leaf $k$ of leaf set $d_{un}$ *captures* $x_n$ . In our notation, all the data captured by a prefix's leaves are also captured by the prefix itself.

Let $\beta$ be a set of leaves. We define $\text{cap}(x_n, \beta) = 1$ if a leaf in $\beta$ captures datum $x_n$, and 0 otherwise. For example, let $d$ and $d'$ be leaf sets such that $d' \supset d$, then $d'$ captures all the data that $d$ captures: $\{x_n : \text{cap}(x_n, d)\} \subseteq \{x_n : \text{cap}(x_n, d')\}$.

The normalized support of $\beta$, denoted $\text{supp}(\beta, \mathbf{x})$, is the fraction of data captured by $\beta$:

$$\text{supp}(\beta, \mathbf{x}) = \frac{1}{N} \sum_{n=1}^{N} \text{cap}(x_n, \beta). \tag{3}$$

## 3.1 Objective Function

For a tree $d = (d_{un}, \delta_{un}, d_{\text{split}}, \delta_{\text{split}}, K, H_d)$, we define its objective function as a combination of the misclassification error and a sparsity penalty on the number of leaves:

$$R(d, \mathbf{x}, \mathbf{y}) = \ell(d, \mathbf{x}, \mathbf{y}) + \lambda H_d. \tag{4}$$

$R(d, \mathbf{x}, \mathbf{y})$ is a regularized empirical risk. The loss $\ell(d, \mathbf{x}, \mathbf{y})$ is the misclassification error of $d$, *i.e.*, the fraction of training data with incorrectly predicted labels. $H_d$ is the number of leaves in the tree $d$. $\lambda H_d$ is a regularization term that penalizes bigger trees. Statistical learning theory provides guarantees for this problem; minimizing the loss subject to a (soft or hard) constraint on model size leads to a low upper bound on test error from the Occham's Razor Bound.

## 3.2 Optimization Framework

We minimize the objective function based on a branch-and-bound framework. We propose a series of specialized bounds that work together to eliminate a large part of the search space. These bounds are discussed in detail in the following paragraphs. Proofs are in the supplementary materials.

Some of our bounds could be adapted directly from CORELS [2], namely these two:
**(Hierarchical objective lower bound)** Lower bounds of a parent tree also hold for every child tree of that parent (§3.3, Theorem 3.1). **(Equivalent points bound)** For a given dataset, if there are multiple samples with exactly the same features but different labels, then no matter how we build our classifier, we will always make mistakes. The lower bound on the number of mistakes is therefore the number of such samples with minority class labels (§B, Theorem B.2).

Some of our bounds adapt from CORELS [1] with minor changes: **(Objective lower bound with one-step lookahead)** With respect to the number of leaves, if a tree does not achieve enough accuracy, we can prune all child trees of it (§3.3, Lemma 3.2). **(A priori bound on the number of leaves)** For an optimal decision tree, we provide an *a priori* upper bound on the maximum number of leaves (§C, Theorem C.3). **(Lower bound on node support)** For an optimal tree, the support traversing through each internal node must be at least $2\lambda$ (§3.4, Theorem 3.3).

Some of our bounds are distinct from CORELS, because they are only relevant to trees and not to lists: **(Lower bound on incremental classification accuracy)** Each split must result in sufficient reduction of the loss. Thus, if the loss reduction is less than or equal to the regularization term, we should still split, and we have to further split at least one of the new child leaves to search for the optimal tree (§3.4, Theorem 3.4). **(Leaf permutation bound)** We need to consider only one

permutation of leaves in a tree; we do not need to consider other permutations (explained in §E, Corollary E.1). **(Leaf accurate support bound)** For each leaf in an optimal decision tree, the number of correctly classified samples must be above a threshold. (§3.4, Theorem 3.5). The supplement contains an additional set of bounds on the number of remaining tree evaluations.

## 3.3  Hierarchical Objective Lower Bound

The loss can be decomposed into two parts corresponding to the unchanged leaves and the leaves to be split: $\ell(d, \mathbf{x}, \mathbf{y}) \equiv \ell_p(d_{un}, \delta_{un}, \mathbf{x}, \mathbf{y}) + \ell_q(d_{\text{split}}, \delta_{\text{split}}, \mathbf{x}, \mathbf{y})$, where $d_{un} = (p_1, \ldots, p_K)$, $\delta_{un} = (\hat{y}_1^{(\text{leaf})}, \ldots, \hat{y}_K^{(\text{leaf})})$, $d_{\text{split}} = (p_{K+1}, \ldots, p_{H_d})$ and $\delta_{\text{split}} = (\hat{y}_{K+1}^{(\text{leaf})}, \ldots, \hat{y}_{H_d}^{(\text{leaf})})$; $\ell_p(d_{un}, \delta_{un}, \mathbf{x}, \mathbf{y}) = \frac{1}{N} \sum_{n=1}^{N} \sum_{k=1}^{K} \text{cap}(x_n, p_k) \wedge \mathbb{1}[\hat{y}_k^{(\text{leaf})} \neq y_n]$ is the proportion of data in the unchanged leaves that are misclassified, and $\ell_p(d_{\text{split}}, \delta_{\text{split}}, \mathbf{x}, \mathbf{y}) = \frac{1}{N} \sum_{n=1}^{N} \sum_{k=K+1}^{H_d} \text{cap}(x_n, p_k) \wedge \mathbb{1}[\hat{y}_k^{(\text{leaf})} \neq y_n]$ is the proportion of data in the leaves we are going to split that are misclassified. We define a lower bound $b(d_{un}, \mathbf{x}, \mathbf{y})$ on the objective by leaving out the latter loss,

$$b(d_{un}, \mathbf{x}, \mathbf{y}) \equiv \ell_p(d_{un}, \delta_{un}, \mathbf{x}, \mathbf{y}) + \lambda H_d \leq R(d, \mathbf{x}, \mathbf{y}), \tag{5}$$

where the leaves $d_{un}$ are kept and the leaves $d_{\text{split}}$ are going to be split. Here, $b(d_{un}, \mathbf{x}, \mathbf{y})$ gives a lower bound on the objective of *any* child tree of $d$.

**Theorem 3.1** (Hierarchical objective lower bound). *Define $b(d_{un}, \mathbf{x}, \mathbf{y}) = \ell_p(d_{un}, \delta_{un}, \mathbf{x}, \mathbf{y}) + \lambda H_d$, as in (5). Define $\sigma(d)$ to be the set of all $d$'s child trees whose unchanged leaves contain $d_{un}$, as in (2). For tree $d = (d_{un}, \delta_{un}, d_{\text{split}}, \delta_{\text{split}}, K, H_d)$ with unchanged leaves $d_{un}$, let $d' = (d'_{un}, \delta'_{un}, d'_{\text{split}}, \delta'_{\text{split}}, K', H_{d'}) \in \sigma(d)$ be any child tree such that its unchanged leaves $d'_{un}$ contain $d_{un}$ and $K' \geq K, H_{d'} \geq H_d$, then $b(d_{un}, \mathbf{x}, \mathbf{y}) \leq R(d', \mathbf{x}, \mathbf{y})$.*

Consider a sequence of trees, where each tree is the parent of the following tree. In this case, the lower bounds of these trees increase monotonically, which is amenable to branch-and-bound. We illustrate our framework in Algorithm 1 in Supplement A. According to Theorem 3.1, we can hierarchically prune the search space. During the execution of the algorithm, we cache the current best (smallest) objective $R^c$, which is dynamic and monotonically decreasing. In this process, when we generate a tree whose unchanged leaves $d_{un}$ correspond to a lower bound satisfying $b(d_{un}, \mathbf{x}, \mathbf{y}) \geq R^c$, according to Theorem 3.1, we do not need to consider *any* child tree $d' \in \sigma(d)$ of this tree whose $d'_{un}$ contains $d_{un}$.

Based on Theorem 3.1, we describe a consequence in Lemma 3.2.

**Lemma 3.2** (Objective lower bound with one-step lookahead). *Let $d$ be a $H_d$-leaf tree with a $K$-leaf prefix and let $R^c$ be the current best objective. If $b(d_{un}, \mathbf{x}, \mathbf{y}) + \lambda \geq R^c$, then for any child tree $d' \in \sigma(d)$, its prefix $d'_{un}$ starts with $d_{un}$ and $K' > K, H_{d'} > H_d$, and it follows that $R(d', \mathbf{x}, \mathbf{y}) \geq R^c$.*

This bound tends to be very powerful in practice in pruning the search space, because it states that *even though we might have a tree with unchanged leaves $d_{un}$ whose lower bound $b(d_{un}, \mathbf{x}, \mathbf{y}) \leq R^c$, if $b(d_{un}, \mathbf{x}, \mathbf{y}) + \lambda \geq R^c$, we can still prune all of its child trees.*

## 3.4  Lower Bounds on Node Support and Classification Accuracy

We provide three lower bounds on the fraction of correctly classified data and the normalized support of leaves in any optimal tree. All of them depend on $\lambda$.

**Theorem 3.3** (Lower bound on node support). *Let $d^* = (d_{un}, \delta_{un}, d_{\text{split}}, \delta_{\text{split}}, K, H_{d^*})$ be any optimal tree with objective $R^*$, i.e., $d^* \in \operatorname{argmin}_d R(d, \mathbf{x}, \mathbf{y})$. For an optimal tree, the support traversing through each internal node must be at least $2\lambda$. That is, for each child leaf pair $p_k, p_{k+1}$ of a split, the sum of normalized supports of $p_k, p_{k+1}$ should be no less than twice the regularization parameter, i.e., $2\lambda$,*

$$2\lambda \leq \text{supp}(p_k, \mathbf{x}) + \text{supp}(p_{k+1}, \mathbf{x}). \tag{6}$$

Therefore, for a tree $d$, if any of its internal nodes capture less than a fraction $2\lambda$ of the samples, it cannot be an optimal tree, even if $b(d_{un}, \mathbf{x}, \mathbf{y}) < R^*$. None of its child trees would be an optimal tree either. Thus, after evaluating $d$, we can prune tree $d$.

**Theorem 3.4** (Lower bound on incremental classification accuracy). *Let $d^* = (d_{un}, \delta_{un}, d_{\text{split}}, \delta_{\text{split}}, K, H_{d^*})$ be any optimal tree with objective $R^*$, i.e., $d^* \in \operatorname{argmin}_d R(d, \mathbf{x}, \mathbf{y})$. Let $d^*$*

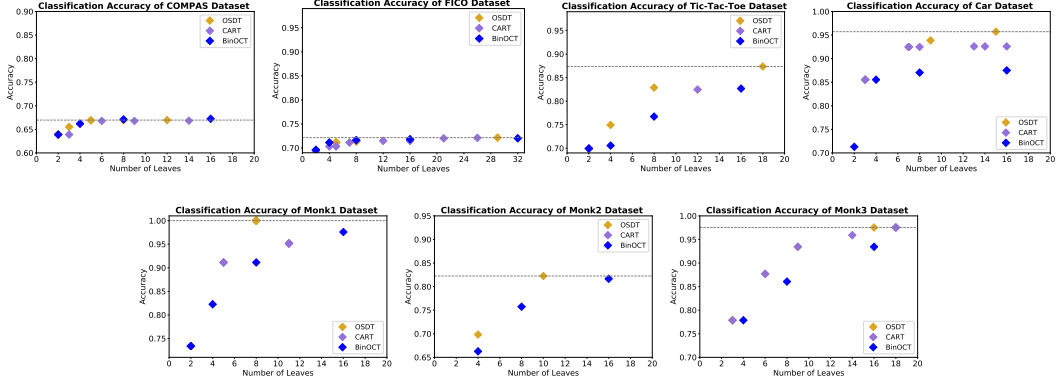

Figure 1: Training accuracy of OSDT, CART, BinOCT on different datasets (time limit: 30 minutes). Horizontal lines indicate the accuracy of the best OSDT tree. On most datasets, all trees of BinOCT and CART are below this line.

*have leaves $d_{un} = (p_1, \ldots, p_{H_{d^*}})$ and labels $\delta_{un} = (\hat{y}_1^{(leaf)}, \ldots, \hat{y}_{H_{d^*}}^{(leaf)})$. For each leaf pair $p_k, p_{k+1}$ with corresponding labels $\hat{y}_k^{(leaf)}, \hat{y}_{k+1}^{(leaf)}$ in $d^*$ and their parent node (the leaf in the parent tree) $p_j$ and its label $\hat{y}_j^{(leaf)}$, define $a_k$ to be the incremental classification accuracy of splitting $p_j$ to get $p_k, p_{k+1}$:*

$$a_k \equiv \frac{1}{N} \sum_{n=1}^{N} \{\mathrm{cap}(x_n, p_k) \wedge \mathbb{1}[\hat{y}_k^{(leaf)} = y_n] + \mathrm{cap}(x_n, p_{k+1}) \wedge \mathbb{1}[\hat{y}_{k+1}^{(leaf)} = y_n] - \mathrm{cap}(x_n, p_j) \wedge \mathbb{1}[\hat{y}_j^{(leaf)} = y_n]\}. \quad (7)$$

*In this case, $\lambda$ provides a lower bound, $\lambda \leq a_k$.*

Thus, when we split a leaf of the parent tree, if the incremental fraction of data that are correctly classified after this split is less than a fraction $\lambda$, we need to further split at least one of the two child leaves to search for the optimal tree. Thus, we apply Theorem 3.3 when we split the leaves. We need only split leaves whose normalized supports are no less than $2\lambda$. We apply Theorem 3.4 when constructing the trees. For every new split, we check the incremental accuracy for this split. If it is less than $\lambda$, we further split at least one of the two child leaves. Both Theorem 3.3 and Theorem 3.4 are bounds for pairs of leaves. We give a bound on a single leaf's classification accuracy in Theorem 3.5.

**Theorem 3.5** (Lower bound on classification accuracy). *Let $d^* = (d_{un}, \delta_{un}, d_{\mathrm{split}}, \delta_{\mathrm{split}}, K, H_{d^*})$ be any optimal tree with objective $R^*$, i.e., $d^* \in \mathrm{argmin}_d R(d, \mathbf{x}, \mathbf{y})$. For each leaf $(p_k, \hat{y}_k^{(leaf)})$ in $d^*$, the fraction of correctly classified data in leaf $k$ should be no less than $\lambda$,*

$$\lambda \leq \frac{1}{N} \sum_{n=1}^{N} \mathrm{cap}(x_n, p_k) \wedge \mathbb{1}[\hat{y}_k^{(leaf)} = y_n]. \quad (8)$$

*Thus, in a leaf we consider extending by splitting on a particular feature, if that proposed split leads to less than $\lambda$ correctly classified data going to either side of the split, then this split can be excluded, and we can exclude that feature anywhere further down the tree extending that leaf.*

W

### 3.5 Incremental Computation

Much of our implementation effort revolves around exploiting incremental computation, designing data structures and ordering of the worklist. Together, these ideas save >97% execution time. We provide the details of our implementation in the supplement.

## 4 Experiments

We address the following questions through experimental analysis: (1) Do existing methods achieve optimal solutions, and if not, how far are they from optimal? (2) How fast does our algorithm converge given the hardness of the problem it is solving? (3) How much does each of the bounds contribute to the performance of our algorithm? (4) What do optimal trees look like?

The results of the per-bound performance and memory improvement experiment (Table 2 in the supplement) were run on a $m5a.4xlarge$ instance of AWS's Elastic Compute Cloud (EC2). The instance has 16 2.5GHz virtual CPUs (although we run single-threaded on a single core) and 64 GB of RAM. All other results were run on a personal laptop with a 2.4GHz i5-8259U processor and 16GB of RAM.

We used 7 datasets: Five of them are from the UCI Machine Learning Repository [8], (Tic Tac Toe, Car Evaluation, Monk1, Monk2, Monk3). The other two datasets are the ProPublica recidivism data set [12] and the Fair Isaac (FICO) credit risk dataset [9]. We predict which individuals are arrested within two years of release ($N = 7, 215$) on the recidivism data set and whether an individual will default on a loan for the FICO dataset.

*Accuracy and optimality:* We tested the accuracy of our algorithm against baseline methods CART and BinOCT [20]. BinOCT is the most recent publicly available method for learning optimal classification trees and was shown to outperform other previous methods. As far as we know, there is no public code for most of the other relevant baselines, including [5, 6, 16]. One of these methods, OCT [5], reports that CART often dominates their performance (see Fig. 4 and Fig. 5 in their paper). Our models can never be worse than CART's models even if we stop early, because in our implementation, we use the objective value of CART's solution as a warm start to the objective value of the current best. Figure 1 shows the training accuracy on each dataset. The time limits for both BinOCT and our algorithm are set to be 30 minutes.

*Main results:* (i) We can now evaluate how close to optimal other methods are (and they are often close to optimal or optimal). (ii) Sometimes, the baselines are *not* optimal. Recall that BinOCT searches only for the optimal tree *given the topology of the complete binary tree of a certain depth.* This restriction on the topology massively reduces the search space so that BinOCT runs quickly, but in exchange, it misses optimal sparse solutions that our method finds. (iii) Our method is *fast.* Our method runs on only one thread (we have not yet parallelized it) whereas BinOCT is highly optimized; it makes use of eight threads. Even with BinOCT's 8-thread parallelism, our method is competitive.

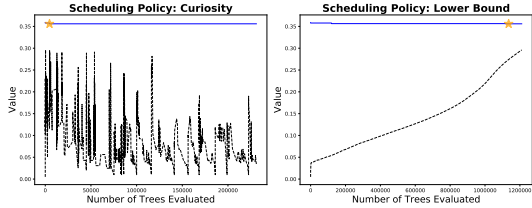

Figure 2: Example OSDT execution traces (COMPAS data, $\lambda = 0.005$). Lines are the objective value and dashes are the lower bound for OSDT. For each scheduling policy, the time to optimum and optimal objective value are marked with a star.

*Convergence:* Figure 2 illustrates the behavior of OSDT for the ProPublica COMPAS dataset with $\lambda = 0.005$, for two different scheduling policies (curiosity and lower bound, see supplement). The charts show how the current best objective value $R^c$ and the lower bound $b(d_{un}, \mathbf{x}, \mathbf{y})$ vary as the algorithm progresses. When we schedule using the lower bound, the lower bounds of evaluated trees increase monotonically, and OSDT certifies optimality only when the value of the lower bound becomes large enough that we can prune the remaining search space or when the queue is empty, whichever is reached earlier. Using curiosity, OSDT finds the optimal tree much more quickly than when using the lower bound.

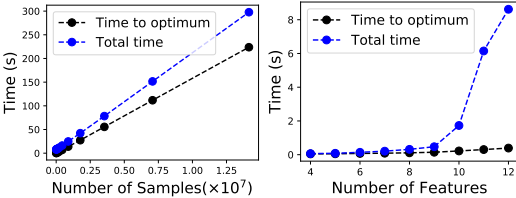

(a) This is based on all the 12 features

(b) The 4 features are those in Figure 4

Figure 3: Scalability with respect to number of samples and number of features using (multiples of) the ProPublica data set. ($\lambda = 0.005$). Note that all these executions include the 4 features of the optimal tree, and the data size are increased by duplicating the whole data set multiple times.

*Scalability:* Figure 3 shows the scalability of OSDT with respect to the number of samples and the number of features. Runtime can theoretically grow exponentially with the number of features. However, as we add extra features that differ from those in the optimal tree, we can reach the optimum more quickly, because we are able to prune the search space more efficiently as the number of extra features grows. For example, with 4 features, it spends about 75% of the runtime to reach the optimum; with 12 features, it takes about 5% of the runtime to reach the optimum.

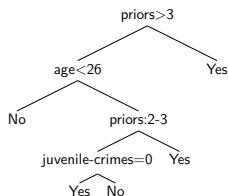

Figure 4: An optimal decision tree generated by OSDT on the COMPAS dataset. ($\lambda = 0.005$, accuracy: 66.90%)

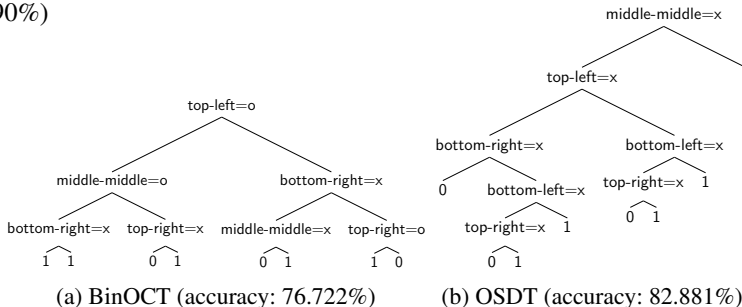

(a) BinOCT (accuracy: 76.722%)     (b) OSDT (accuracy: 82.881%)

Figure 5: Eight-leaf decision trees generated by BinOCT and OSDT on the Tic-Tac-Toe data. Trees of BinOCT must be complete binary trees, while OSDT can generate binary trees of any shape.

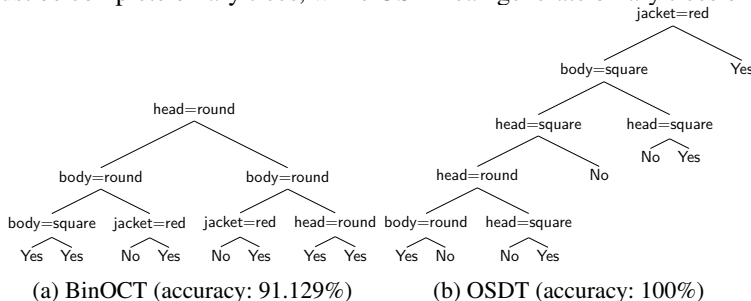

(a) BinOCT (accuracy: 91.129%)     (b) OSDT (accuracy: 100%)

Figure 6: Decision trees generated by BinOCT and OSDT on the Monk1 dataset. The tree generated by BinOCT includes two useless splits (the left and right splits), while OSDT can avoid this problem. BinOCT is 91% accurate, OSDT is 100% accurate.

*Ablation experiments:* Appendix I shows that the lookahead and equivalent points bounds are, by far, the most significant of our bounds, reducing time to optimum by at least two orders of magnitude and reducing memory consumption by more than one order of magnitude.

*Trees:* We provide illustrations of the trees produced by OSDT and the baseline methods in Figures 4, 5 and 6. OSDT generates trees of any shape, and our objective penalizes trees with more leaves, thus it never introduces splits that produce a pair of leaves with the same label. In contrast, BinOCT trees are always complete binary trees of a given depth. This limitation on the tree shape can prevent BinOCT from finding the globally optimal tree. In fact, BinOCT often produces useless splits, leading to trees with more leaves than necessary to achieve the same accuracy.

*Additional experiments:* It is well-established that simpler models such as small decision trees generalize well; a set of cross-validation experiments is in the supplement demonstrating this.

**Conclusion:** Our work shows the possibility of optimal (or provably near-optimal) sparse decision trees. It is the first work to balance the accuracy and the number of leaves optimally in a practical amount of time. We have reason to believe this framework can be extended to much larger datasets. Theorem F.1 identifies a key mechanism for scaling these algorithms up. It suggests a bound stating that highly correlated features can substitute for each other, leading to similar model accuracies. Applications of this bound allow for the elimination of features throughout the entire execution, allowing for more aggressive pruning. Our experience to date shows that by supporting such bounds with the right data structures can potentially lead to dramatic increases in performance and scalability.

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
