[Supplementary Material 1 · OSDT_NIPS_Supplement.pdf]

# Optimal Sparse Decision Trees: Supplementary Material

## A  Branch and Bound Algorithm

Algorithm 1 shows the structure of our approach.

## B  Equivalent Points Bound

When multiple observations captured by a leaf in $d_{\text{split}}$ have identical features but opposite labels, then no tree, including those that extend $d_{\text{split}}$, can correctly classify all of these observations. The number of misclassifications must be at least the minority label of the equivalent points.

For data set $\{(x_n, y_n)\}_{n=1}^N$ and a set of features $\{s_m\}_{m=1}^M$, we define a set of samples to be equivalent if they have exactly the same feature values, *i.e.*, $(x_i, y_i)$ and $(x_j, y_j)$ are equivalent if $\frac{1}{M}\sum_{m=1}^M \mathbb{1}[\text{cap}(x_i, s_m) = \text{cap}(x_j, s_m)] = 1$. Note that a data set consists of multiple sets of equivalent points; let $\{e_u\}_{u=1}^U$ enumerate these sets. For each observation $x_i$, it belongs to a equivalent points set $e_u$. We denote the fraction of data with the minority label in set $e_u$ as $\theta(e_u)$, *e.g.*, let $e_u = \{x_n : \forall m \in [M],\ \mathbb{1}[\text{cap}(x_n, s_m) = \text{cap}(x_i, s_m)]\}$, and let $q_u$ be the minority class label among points in $e_u$, then

$$\theta(e_u) = \frac{1}{N}\sum_{n=1}^N \mathbb{1}[x_n \in e_u]\, \mathbb{1}[y_n = q_u]. \tag{9}$$

We can combine the equivalent points bound with other bounds to get a tighter lower bound on the objective function. As the experimental results demonstrate in §4, there is sometimes a substantial reduction of the search space after incorporating the equivalent points bound. We propose a general equivalent points bound in Proposition B.1. We incorporate it into our framework by proposing the specific equivalent points bound in Theorem B.2.

**Proposition B.1** (General equivalent points bound). *Let $d = (d_{un}, \delta_{un}, d_{\text{split}}, \delta_{\text{split}}, K, H)$ be a tree, then $R(d, \mathbf{x}, \mathbf{y}) \geq \sum_{u=1}^U \theta(e_u) + \lambda H$.*

---

**Algorithm 1** Branch-and-bound for learning optimal decision trees.

---

**Input:** Objective function $R(d, \mathbf{x}, \mathbf{y})$, objective lower bound $b(d_{un}, \mathbf{x}, \mathbf{y})$, set of features $S = \{s_m\}_{m=1}^M$, training data $(\mathbf{x}, \mathbf{y}) = \{(x_n, y_n)\}_{n=1}^N$, initial best known tree $d^0$ with objective $R^0 = R(d^0, \mathbf{x}, \mathbf{y})$; $d^0$ could be obtained as output from another (approximate) algorithm, otherwise, $(d^0, R^0) = (\text{null}, 1)$ provides reasonable default values. The initial value of $\delta_{\text{split}}$ is the majority label of the whole dataset.
**Output:** Provably optimal decision tree $d^*$ with minimum objective $R^*$

$(d^c, R^c) \leftarrow (d^0, R^0)$            ▷ Initialize best tree and objective
$Q \leftarrow \text{queue}(\,[\,((), (), (), \delta_{\text{split}}, 0, 0)\,]\,)$      ▷ Initialize queue with empty tree
**while** $Q$ not empty **do**             ▷ Stop when queue is empty
     $d = (d_{un}, \delta_{un}, d_{\text{split}}, \delta_{\text{split}}, K, H) \leftarrow Q.\text{pop}(\,)$    ▷ Remove tree $d$ from the queue
     **if** $b(d_{un}, \mathbf{x}, \mathbf{y}) < R^c$ **then**         ▷ **Bound**: Apply Theorem 3.1
         $R \leftarrow R(d, \mathbf{x}, \mathbf{y})$         ▷ Compute objective of tree $d$
         **if** $R < R^c$ **then**         ▷ Update best tree and objective
            $(d^c, R^c) \leftarrow (d, R)$
         **end if**
         **for** every possible combination of features to split $d_{\text{split}}$ **do**
                              ▷ **Branch**: Enqueue $d_{un}$'s children
            split $d_{\text{split}}$ and get new leaves $d_{\text{new}}$
            **for** each possible subset $d'_{\text{split}}$ of $d_{\text{new}}$ **do**
                $d'_{un} = d_{un} \cup (d_{\text{new}} \setminus d'_{\text{split}})$
                $Q.\text{push}(\,(d'_{un}, \delta'_{un}, d'_{\text{split}}, \delta'_{\text{split}}, K', H')\,)$
            **end for**
         **end for**
     **end if**
**end while**
$(d^*, R^*) \leftarrow (d^c, R^c)$              ▷ Identify provably optimal solution

---

Recall that in our lower bound $b(d_{un}, \mathbf{x}, \mathbf{y})$ in (5), we leave out the misclassification errors of leaves we are going to split $\ell_0(d_{\text{split}}, \delta_{\text{split}}, \mathbf{x}, \mathbf{y})$ from the objective $R(d, \mathbf{x}, \mathbf{y})$. Incorporating the equivalent points bound in Theorem B.2, we obtain a tighter bound on our objective because we now have a tighter lower bound on the misclassification errors of leaves we are going to split, $0 \le b_0(d_{\text{split}}, \mathbf{x}, \mathbf{y}) \le \ell_0(d_{\text{split}}, \delta_{\text{split}}, \mathbf{x}, \mathbf{y})$.

**Theorem B.2** (Equivalent points bound). *Let $d$ be a tree with leaves $d_{un}, d_{\text{split}}$ and lower bound $b(d_{un}, \mathbf{x}, \mathbf{y})$, then for any tree $d' \in \sigma(d)$ whose prefix $d'_{un} \supseteq d_{un}$,*

$$R(d', \mathbf{x}, \mathbf{y}) \ge b(d_{un}, \mathbf{x}, \mathbf{y}) + b_0(d_{\text{split}}, \mathbf{x}, \mathbf{y}), \quad \text{where} \tag{10}$$

$$b_0(d_{\text{split}}, \mathbf{x}, \mathbf{y}) = \frac{1}{N} \sum_{u=1}^{U} \sum_{n=1}^{N} \text{cap}(x_n, d_{\text{split}}) \wedge \mathbb{1}[x_n \in e_u]\, \mathbb{1}[y_n = q_u]. \tag{11}$$

# C  Upper Bounds on Number of Leaves

During the branch-and-bound execution, the current best objective $R^c$ implies an upper bound on the maximum number of leaves for those trees we still need to consider.

**Theorem C.1** (Upper bound on the number of leaves). *For a dataset with $M$ features, consider a state space of all trees. Let $L(d)$ be the number of leaves of tree $d$ and let $R^c$ be the current best objective. For all optimal trees $d^* \in \text{argmin}_d R(d, \mathbf{x}, \mathbf{y})$*

$$L(d^*) \le \min\left(\lfloor R^c/\lambda \rfloor, 2^M\right), \tag{12}$$

*where $\lambda$ is the regularization parameter.*

**Corollary C.2** (A priori upper bound on the number of leaves). *For all optimal trees $d^* \in \text{argmin}_d R(d, \mathbf{x}, \mathbf{y})$,*

$$L(d^*) \le \min\left(\lfloor 1/2\lambda \rfloor, 2^M\right). \tag{13}$$

For any particular tree $d$ with unchanged leaves $d_{un}$, we can obtain potentially tighter upper bounds on the number of leaves for all its child trees whose unchanged leaves include $d_{un}$.

**Theorem C.3** (Parent-specific upper bound on the number of leaves). *Let $d = (d_{un}, \delta_{un}, d_{\text{split}}, \delta_{\text{split}}, K, H)$ be a tree, let $d' = (d'_{un}, \delta'_{un}, d'_{\text{split}}, \delta'_{\text{split}}, K', H') \in \sigma(d)$ be any child tree such that $d'_{un} \supseteq d_{un}$, and let $R^c$ be the current best objective. If $d'_{un}$ has lower bound $b(d'_{un}, \mathbf{x}, \mathbf{y}) < R^c$, then*

$$H' < \min\left(H + \left\lfloor \frac{R^c - b(d_{un}, \mathbf{x}, \mathbf{y})}{\lambda} \right\rfloor, 2^M\right). \tag{14}$$

Theorem C.3 can be viewed as a generalization of the one-step lookahead bound (Lemma 3.2). This is because we can view (14) as a bound on $H' - H$, which provides an upper bound on the number of remaining splits we may need, based on the best tree we already have evaluated.

# D  Upper Bounds on Number of Tree Evaluations

In this section, based on the upper bounds on the number of leaves from §C, we give corresponding upper bounds on the number of tree evaluations made by Algorithm 1. First, in Theorem D.1, based on information about the state of Algorithm 1's execution, we calculate, for any given execution state, upper bounds on the number of additional tree evaluations needed for the execution to complete. We define the number of *remaining tree evaluations* as the number of trees that are currently in, or will be inserted into, the queue. We evaluate the number of tree evaluations based on current execution information of the current best objective $R^c$ and the trees in the queue $Q$ of Algorithm 1.

**Theorem D.1** (Upper bound on number of remaining tree evaluations). *Consider the state space of all possible leaves formed from a set of $M$ features, and consider Algorithm 1 at a particular instant during execution. Denote the current best objective as $R^c$, the queue as $Q$, and the size of prefix $d_{un}$ as $L(d_{un})$. Denoting the number of remaining prefix evaluations as $\Gamma(R^c, Q)$, the bound is:*

$$\Gamma(R^c, Q) \le \sum_{d_{un} \in Q} \sum_{k=0}^{f(d_{un})} \frac{(3^M - L(d_{un}))!}{(3^M - L(d_{un}) - k)!}, \tag{15}$$

$$\text{where} \quad f(d_{un}) = \min\left(\left\lfloor \frac{R^c - b(d_{un}, \mathbf{x}, \mathbf{y})}{\lambda} \right\rfloor, 3^M - L(d_{un})\right). \tag{16}$$

The corollary below is a naïve upper bound on the total number of tree evaluations during the process of Algorithm 1's execution. It does not use algorithm execution state to bound the size of the search space like Theorem D.1, and it relies only on the number of features and the regularization parameter $\lambda$.

**Corollary D.2** (Upper bound on the total number of tree evaluations)**.** *Define $\Gamma_{\text{tot}}(S)$ to be the total number of trees evaluated by Algorithm 1, given the state space of all possible leaves formed from a set $S$ of $M$ features. For any set $S$ of all leaves formed of $M$ features,*

$$\Gamma_{\text{tot}}(S) \leq \sum_{k=0}^{K} \frac{3^M!}{(3^M - k)!}, \text{ where } K = \min(\lfloor 1/2\lambda \rfloor, 2^M).$$

# E    Permutation Bound

If two trees are composed of the same leaves, *i.e.*, they contain the same conjunctions of features up to a permutation, then they classify all the data in the same way and their child trees are also permutations of each other. Therefore, if we already have all children from one permutation of a tree, then there is no benefit to considering child trees generated from a different permutation.

**Corollary E.1** (Leaf Permutation bound)**.** *Let $\pi$ be any permutation of $\{1, \ldots, H\}$, Let $d = (d_{un}, \delta_{un}, d_{\text{split}}, \delta_{\text{split}}, K, H)$ and $D = (D_{un}, \Delta_{un}, D_{\text{split}}, \Delta_{\text{split}}, K, H)$ denote trees with leaves $(p_1, \ldots, p_H)$ and $D_{un} = (p_{\pi(1)}, \ldots, p_{\pi(H)})$, respectively, i.e., the leaves in $D$ correspond to a permutation of the leaves in $d$. Then the objective lower bounds of $d$ and $D$ are the same and their child trees correspond to permutations of each other.*

Therefore, if two trees have the same leaves, up to a permutation, according to Corollary E.1, either of them can be pruned. We call this symmetry-aware pruning. In Section §E.1, we demonstrate how this helps to save computation.

## E.1    Upper bound on tree evaluations with symmetry-aware pruning

Here we give an upper bound on the total number of tree evaluations based on symmetry-aware pruning (§E). For every subset of $K$ leaves, there are $K!$ leaf sets equivalent up to permutation. Thus, symmetry-aware pruning dramatically reduces the search space by considering only one of them. This effects the execution of Algorithm 1's breadth-first search. With symmetry-aware pruning, when it evaluates trees of size $K$, for each set of trees equivalent up to a permutation, it keeps only a single tree.

**Theorem E.2** (Upper bound on tree evaluations with symmetry-aware pruning)**.** *Consider a state space of all trees formed from a set $S$ of $3^M$ leaves where $M$ is the number of features (the 3 options correspond to having the feature's value be 1, having its value be 0, or not including the feature), and consider the branch-and-bound algorithm with symmetry-aware pruning. Define $\Gamma_{\text{tot}}(S)$ to be the total number of prefixes evaluated. For any set $S$ of $3^M$ leaves,*

$$\Gamma_{\text{tot}}(S) \leq 1 + \sum_{k=1}^{K} N_k + C(M, k) - P(M, k), \tag{17}$$

*where $K = \min(\lfloor 1/2\lambda \rfloor, 2^M)$, $N_k$ is defined in* (1)*.*

*Proof.* By Corollary C.2, $K \equiv \min(\lfloor 1/2\lambda \rfloor, 2^M)$ gives an upper bound on the number of leaves of any optimal tree. The algorithm begins by evaluating the empty tree, followed by $M$ trees of depth $k = 1$, then $N_2 = \sum_{n_0=1}^{1} \sum_{n_1=1}^{2^{n_0}} M \times \binom{2^{n_0}}{n_1}(M-1)^{n_1}$ trees of depth $k = 2$. Before proceeding to length $k = 3$, we keep only $N_2 + C(M, 2) - P(M, 2)$ trees of depth $k = 2$, where $N_k$ is defined in (1), $P(M, k)$ denotes the number of $k$-permutations of $M$ and $C(M, k)$ denotes the number of $k$-combinations of $M$. Now, the number of length $k = 3$ prefixes we evaluate is $N_3 + C(M, 3) - P(M, 3)$. Propagating this forward gives (17). $\qquad \square$

Pruning based on permutation symmetries thus yields significant computational savings of $\sum_{k=1}^{K} P(M, k) - C(M, k)$. For example, when $M = 10$ and $K = 5$, the number reduced due to symmetry-aware pruning is about 35463. If $M = 20$ and $K = 10$, the number of evaluations is reduced by about $7.36891 \times 10^{11}$.

# F    Similar Support Bound

Here, we present the similar support bound to deal with similar trees. Let us say we are given two trees that are the same except that one internal node is split by a different feature, where this second feature is similar to the first tree's feature. By the similar support bound, if we know that one of these trees and all its child trees are worse (beyond a margin) than the current best tree, we can prune the other one and all of its child trees.

**Theorem F.1** (Similar support bound). *Define* $d = (d_{un}, \delta_{un}, d_{\text{split}}, \delta_{\text{split}}, K, H)$ *and* $D = (D_{un}, \Delta_{un}, D_{\text{split}}, \Delta_{\text{split}}, K, H)$ *to be two trees which are exactly the same but one internal node split by different features. Let* $f_1, f_2$ *be the features used to split that node in* $d$ *and* $D$ *respectively. Let* $t_1, t_2$ *be the left subtree and the right subtree under the node* $f_1$ *in* $d$, *and let* $T_1, T_2$ *be the left subtree and the right subtree under the node* $f_2$ *in* $D$. *Denote the normalized support of data captured by only one of* $t_1$ *and* $T_1$ *as* $\omega$, *i.e.,*

$$\omega \equiv \frac{1}{N} \sum_{n=1}^{N} [\neg \operatorname{cap}(x_n, t_1) \wedge \operatorname{cap}(x_n, T_1) + \operatorname{cap}(x_n, t_1) \wedge \neg \operatorname{cap}(x_n, T_1)]. \tag{18}$$

*The difference between the two trees' objectives is bounded by* $\omega$ *as the following:*

$$\omega \geq R(d, \mathbf{x}, \mathbf{y}) - R(D, \mathbf{x}, \mathbf{y}) \geq -\omega, \tag{19}$$

*where* $R(d, \mathbf{x}, \mathbf{y})$ *is objective of* $d$ *and* $R(D, \mathbf{x}, \mathbf{y})$ *is the objective of* $D$. *Then, we have*

$$\omega \geq \min_{d^\dagger \in \sigma(d)} R(d^\dagger, \mathbf{x}, \mathbf{y}) - \min_{D^\dagger \in \sigma(D_{un})} R(D^\dagger, \mathbf{x}, \mathbf{y}) \geq -\omega. \tag{20}$$

*Proof.* The difference between the objectives of $d$ and $D$ is maximized when one of them correctly classifies all the data corresponding to $\omega$ but the other misclassifies all of them. Therefore,

$$\omega \geq R(d, \mathbf{x}, \mathbf{y}) - R(D, \mathbf{x}, \mathbf{y}) \geq -\omega.$$

Let $d^*$ be the best child tree of $d$, *i.e.*, $R(d^*, \mathbf{x}, \mathbf{y}) = \min_{d^\dagger \in \sigma(d)} R(d^\dagger, \mathbf{x}, \mathbf{y})$, and let $D' \in \sigma(D_{un})$ be its counterpart which is exactly the same but one internal node split by a different feature. Because $R(D', \mathbf{x}, \mathbf{y}) \geq \min_{D^\dagger \in \sigma(D_{un})} R(D^\dagger, \mathbf{x}, \mathbf{y})$,

$$\min_{d^\dagger \in \sigma(d)} R(d^\dagger, \mathbf{x}, \mathbf{y}) = R(d^*, \mathbf{x}, \mathbf{y}) \geq R(D', \mathbf{x}, \mathbf{y}) - \omega$$

$$\geq \min_{D^\dagger \in \sigma(D_{un})} R(D^\dagger, \mathbf{x}, \mathbf{y}) - \omega. \tag{21}$$

Similarly, $\min_{D^\dagger \in \sigma(D_{un})} R(D^\dagger, \mathbf{x}, \mathbf{y}) + \omega \geq \min_{d^\dagger \in \sigma(d)} R(d^\dagger, \mathbf{x}, \mathbf{y})$. $\qquad\square$

With the similar support bound, for two trees $d$ and $D$ as above, if we already know $D$ and all its child trees cannot achieve a better tree than the current optimal one, and in particular, we assume we know that $\min_{D^\dagger \in \sigma(D_{un})} R(D^\dagger, \mathbf{x}, \mathbf{y}) \geq R^c + \omega$, we can then also prune $d$ and all of its child trees, because

$$\min_{d^\dagger \in \sigma(d)} R(d^\dagger, \mathbf{x}, \mathbf{y}) \geq \min_{D^\dagger \in \sigma(D_{un})} R(D^\dagger, \mathbf{x}, \mathbf{y}) - \omega \geq R^c. \tag{22}$$

# G Implementation

We implement a series of data structures designed to support incremental computation.

## G.1 Data Structure of Leaf and Tree

First, we store bounds and intermediate results for both full trees and the individual leaves to support the incremental computation of the lower bound and the objective. As a global statistic, we maintain the best (minimum) observed value of the objective function and the corresponding tree. As each leaf in a tree represents a set of clauses, each leaf stores a bit-vector representing the set of samples captured by that clause and the prediction accuracy for those samples. From these values in the leaves, we can efficiently compute both the value of the objective for an entire tree and new leaf values for children formed from splitting a leaf.

Specifically, for the data structure of leaf $l$, we store:

- A set of clauses defining the leaf.
- A binary vector of length $N$ (number of data points) indicating whether or not each point is captured by the leaf.
- The number of points captured by the leaf.
- A binary vector of length $M$ (number of features) indicating the set of dead features. In a leaf, a feature is dead if Theorem 3.5 does not hold.
- The lower bound on the leaf misclassification error $b_0(l, \mathbf{x}, \mathbf{y})$, which is defined in (11)
- The label of the leaf.
- The loss of the leaf.

- A boolean indicating whether the leaf is dead. A leaf is dead if Theorem 3.3 does not hold; we never split a dead leaf.

We store additional information for entire trees:

- A set of leaves in the tree.

- The objective.

- The lower bound of the objective.

- A binary vector of length $n_l$ (number of leaves) indicating whether the leaf can be split, that is, this vector records split leaves $d_{\text{split}}$ and unchanged leaves $d_{un}$. The unchanged leaves of a tree are also unchanged leaves in its child trees.

## G.2 Queue

Second, we use a priority queue to order the exploration of the search space. The queue serves as a worklist, with each entry in the queue corresponding to a tree. When an entry is removed from the queue, we use it to generate child trees, incrementally computing the information for the child trees. The ordering of the worklist represents a scheduling policy. We evaluated both structural orderings, e.g., breadth first search and depth first search, and metric-based orderings, e.g., objective function, and its lower bound. Each metric produces a different scheduling policy. We achieve the best performance in runtime and memory consumption using the curiosity metric from CORELS [2], which is the objective's lower bound, divided by the normalized support of its unchanged leaves. For example, relative to using the objective, curiosity reduces runtime by a factor of two and memory consumption by a factor of four.

## G.3 Symmetry-aware Map

Third, to support symmetry-aware pruning from Corollary E.1, we introduce two symmetry aware maps – a LeafCache and a TreeCache. The LeafCache ensures that we only compute values for a particular leave once; the TreeCache ensures we do not create trees equivalent to those we have already explored.

A leaf is a set of clauses, each of which corresponds to an attribute and the value (0,1) of that attribute. As the leaves of a decision tree are mutually exclusive, the data captured by each leaf is insensitive to the order of the leaf's clauses. We encode leaves in a canonical order (sorted by attribute indexes) and use that canonical order as the key into the LeafCache. Each entry in the LeafCache represents all permutations of a set of clauses. We use a Python dictionary to map these keys to the leaf and its cached values. Before creating a leaf object, we first check if we already have that leaf in our map. If not, we create the leaf and insert it into the map. Otherwise, the permutation already exists, so we use the cached copy in the tree we are constructing.

Next, we implement the permutation bound (Corollary E.1) using the TreeCache. The TreeCache contains encodings of all the trees we have evaluated. Like we did for the clauses in the LeafCache, we introduce a canonical order over the leaves in a tree and use that as the key to the TreeCache. If our algorithm produces a tree that is a permutation of a tree we have already evaluated, we need not evaluate it again. Before evaluating a tree, we look it up in the cache. If it's in the cache, we do nothing; if it is not, we compute the bounds for the tree and insert it into the cache.

## G.4 Execution

Now, we illustrate how these data structures support execution of our algorithm. We initialize the algorithm with the current best objective $R^c$ and tree $d^c$. For unexplored trees in the queue, the scheduling policy selects the next tree $d$ to split; we keep removing elements from the queue until the queue is empty. Then, for every possible combination of features to split $d_{\text{split}}$, we construct a new tree $d'$ with incremental calculation of the lower bound $b(d'_{un}, \mathbf{x}, \mathbf{y})$ and the objective $R(d', \mathbf{x}, \mathbf{y})$. If we achieve a better objective $R(d', \mathbf{x}, \mathbf{y})$, i.e., less than the current best objective $R^c$, we update $R^c$ and $d^c$. If the lower bound of the new tree $d'$, combined with the equivalent points bound (Theorem B.2) and the lookahead bound (Theorem 3.2), is less than the current best objective, then we push it into the queue. Otherwise, according to the hierarchical lower bound (Theorem 3.1), no child of $d'$ could possibly have an objective better than $R^c$, which means we do not push $d'$ queue. When there are no more trees to explore, i.e., the queue is empty, we have finished the search of the whole space and output the (provably) optimal tree.

# H Proof of Theorems

## H.1 Proof of Theorem C.1

*Proof.* For an optimal tree $d^*$ with objective $R^*$,

$$\lambda L(d^*) \leq R^* = R(d^*, \mathbf{x}, \mathbf{y}) = \ell(d^*, \mathbf{x}, \mathbf{y}) + \lambda L(d^*) \leq R^c.$$

The maximum possible number of leaves for $d^*$ occurs when $\ell(d^*, \mathbf{x}, \mathbf{y})$ is minimized; therefore this gives bound (12).

For the rest of the proof, let $H^* = L(d^*)$ be the number of leaves of $d^*$. If the current best tree $d^c$ has zero misclassification error, then

$$\lambda H^* \leq \ell(d^*, \mathbf{x}, \mathbf{y}) + \lambda H^* = R(d^*, \mathbf{x}, \mathbf{y}) \leq R^c = R(d^c, \mathbf{x}, \mathbf{y}) = \lambda H,$$

and thus $H^* \leq H$. If the current best tree is suboptimal, *i.e.*, $d^c \notin \operatorname{argmin}_d R(d, \mathbf{x}, \mathbf{y})$, then

$$\lambda H^* \leq \ell(d^*, \mathbf{x}, \mathbf{y}) + \lambda H^* = R(d^*, \mathbf{x}, \mathbf{y}) < R^c = R(d^c, \mathbf{x}, \mathbf{y}) = \lambda H,$$

in which case $H^* < H$, *i.e.*, $H^* \leq H - 1$, since $H$ is an integer. $\square$

## H.2 Proof of Theorem C.3

*Proof.* First, note that $H' \geq H$. Now recall that

$$
\begin{aligned}
b(d_{un}, \mathbf{x}, \mathbf{y}) &= \ell_p(d_{un}, \delta_{un}, \mathbf{x}, \mathbf{y}) + \lambda H \\
&\leq \ell_p(d'_{un}, \delta'_{un}, \mathbf{x}, \mathbf{y}) + \lambda H' = b(d'_{un}, \mathbf{x}, \mathbf{y}),
\end{aligned}
$$

and that $\ell_p(d_{un}, \delta_{un}, \mathbf{x}, \mathbf{y}) \leq \ell_p(d'_{un}, \delta'_{un}, \mathbf{x}, \mathbf{y})$. Combining these bounds and rearranging gives

$$
\begin{aligned}
b(d'_{un}, \mathbf{x}, \mathbf{y}) &= \ell_p(d'_{un}, \delta'_{un}, \mathbf{x}, \mathbf{y}) + \lambda H + \lambda(H' - H) \\
&\geq \ell_p(d_{un}, \delta_{un}, \mathbf{x}, \mathbf{y}) + \lambda H + \lambda(H' - H) \\
&= b(d_{un}, \mathbf{x}, \mathbf{y}) + \lambda(H' - H). \quad (23)
\end{aligned}
$$

Combining (23) with $b(d'_{un}, \mathbf{x}, \mathbf{y}) < R^c$ gives (14). $\square$

## H.3 Proof of Theorem D.1

*Proof.* The number of remaining tree evaluations is equal to the number of trees that are currently in or will be inserted into queue $Q$. For any such tree with unchanged leaves $d_{un}$, Theorem C.3 gives an upper bound on the number of leaves of a tree with unchanged leaves $d'_{un}$ that contains $d_{un}$:

$$L(d'_{un}) \leq \min\left(L(d_{un}) + \left\lfloor \frac{R^c - b(d_{un}, \mathbf{x}, \mathbf{y})}{\lambda} \right\rfloor, 2^M\right) \equiv U(d_{un}).$$

This gives an upper bound on the remaining tree evaluations:

$$\Gamma(R^c, Q) \leq \sum_{d_{un} \in Q} \sum_{k=0}^{U(d_{un}) - L(d_{un})} P(3^M - L(d_{un}), k) \quad (24)$$

$$= \sum_{d_{un} \in Q} \sum_{k=0}^{f(d_{un})} \frac{(3^M - L(d_{un}))!}{(3^M - L(d_{un}) - k)!},$$

where $P(m, k)$ denotes the number of $k$-permutations of $m$. $\square$

## H.4 Proof of Proposition D.2

*Proof.* By Corollary C.2, $K \equiv \min(\lfloor 1/2\lambda \rfloor, 2^M)$ gives an upper bound on the number of leaves of any optimal tree. Since we can think of our problem as finding the optimal selection and permutation of $k$ out of $3^M$ leaves, over all $k \leq K$,

$$\Gamma_{\text{tot}}(S) \leq 1 + \sum_{k=1}^{K} P(3^M, k) = \sum_{k=0}^{K} \frac{3^M!}{(3^M - k)!}.$$

$\square$

## H.5 Proof of Theorem 3.3

*Proof.* Let $d^* = (d_{un}, \delta_{un}, d_{\text{split}}, \delta_{\text{split}}, K, H)$ be an optimal tree with leaves $(p_1, \ldots, p_H)$ and labels $(\hat{y}_1^{(\text{leaf})}, \ldots, \hat{y}_H^{(\text{leaf})})$. Consider the tree $d = (d'_{un}, \delta'_{un}, d'_{\text{split}}, \delta'_{\text{split}}, K', H')$ derived from $d^*$ by deleting a pair of sibling leaves $p_i \to \hat{y}_i^{(\text{leaf})}, p_{i+1} \to \hat{y}_{i+1}^{(\text{leaf})}$ and adding their parent leaf $p_j \to \hat{y}_j^{(\text{leaf})}$, therefore $d'_{un} = (p_1, \ldots, p_{i-1}, p_{i+2}, \ldots, p_H, p_j)$ and $\delta'_{un} = (\hat{y}_1^{(\text{leaf})}, \ldots, \hat{y}_{i-1}^{(\text{leaf})}, \hat{y}_{i+2}^{(\text{leaf})}, \ldots, \hat{y}_H^{(\text{leaf})}, \hat{y}_j^{(\text{leaf})})$.

When $d$ misclassifies half of the data captured by $p_i, p_{i+1}$, while $d^*$ correctly classifies them all, the difference between $d$ and $d^*$ would be maximized, which provides an upper bound:

$$
\begin{aligned}
R(d, \mathbf{x}, \mathbf{y}) &= \ell(d, \mathbf{x}, \mathbf{y}) + \lambda(H - 1) \\
&\leq \ell(d^*, \mathbf{x}, \mathbf{y}) + \text{supp}(p_i, \mathbf{x}) + \text{supp}(p_{i+1}, \mathbf{x}) \\
&\quad - \frac{1}{2}[\text{supp}(p_i, \mathbf{x}) + \text{supp}(p_{i+1}, \mathbf{x})] + \lambda(H - 1) \\
&= R(d^*, \mathbf{x}, \mathbf{y}) + \frac{1}{2}[\text{supp}(p_i, \mathbf{x}) + \text{supp}(p_{i+1}, \mathbf{x})] - \lambda \\
&= R^* + \frac{1}{2}[\text{supp}(p_i, \mathbf{x}) + \text{supp}(p_{i+1}, \mathbf{x})] - \lambda
\end{aligned}
\tag{25}
$$

where $\text{supp}(p_i, \mathbf{x}), \text{supp}(p_i, \mathbf{x})$ is the normalized support of $p_i, p_{i+1}$, defined in (3), and the regularization 'bonus' comes from the fact that $d^*$ has one more leaf than $d$.

Because $d^*$ is the optimal tree, we have $R^* \leq R(d, \mathbf{x}, \mathbf{y})$, which combined with (25) leads to (6). Therefore, for each child leaf pair $p_k, p_{k+1}$ of a split, the sum of normalized supports of $p_k, p_{k+1}$ should be no less than twice the regularization parameter, *i.e.*, $2\lambda$. □

## H.6 Proof of Theorem 3.4

*Proof.* Let $d = (d'_{un}, \delta'_{un}, d'_{\text{split}}, \delta'_{\text{split}}, K', H')$ be the tree derived from $d^*$ by deleting a pair of leaves, $p_i$ and $p_{i+1}$, and adding the their parent leaf, $p_j$. The discrepancy between $d^*$ and $d$ is the discrepancy between $(p_i, p_{i+1})$ and $p_j$: $\ell(d, \mathbf{x}, \mathbf{y}) - \ell(d^*, \mathbf{x}, \mathbf{y}) = a_i$, where $a_i$ is defined in (7). Therefore,

$$
\begin{aligned}
R(d, \mathbf{x}, \mathbf{y}) &= \ell(d, \mathbf{x}, \mathbf{y}) + \lambda(K - 1) = \ell(d^*, \mathbf{x}, \mathbf{y}) + a_i + \lambda(K - 1) \\
&= R(d^*, \mathbf{x}, \mathbf{y}) + a_i - \lambda = R^* + a_i - \lambda.
\end{aligned}
$$

This combined with $R^* \leq R(d, \mathbf{x}, \mathbf{y})$ leads to $\lambda \leq a_i$. □

## H.7 Proof of Theorem 3.5

*Proof.* Let $d = (d'_{un}, \delta'_{un}, d'_{\text{split}}, \delta'_{\text{split}}, K', H')$ be the tree derived from $d^*$ by deleting a pair of leaves, $p_i$ with label $\hat{y}_i^{(\text{leaf})}$ and $p_{i+1}$ with label $\hat{y}_{i+1}^{(\text{leaf})}$, and adding the their parent leaf $p_j$ with label $\hat{y}_j^{(\text{leaf})}$. The discrepancy between $d^*$ and $d$ is the discrepancy between $p_i, p_{i+1}$ and $p_j$: $\ell(d, \mathbf{x}, \mathbf{y}) - \ell(d^*, \mathbf{x}, \mathbf{y}) = a_i$, where we defined $a_i$ in (7). According to Theorem 3.4, $\lambda \leq a_i$ and

$$
\begin{aligned}
\lambda \leq \frac{1}{N} \sum_{n=1}^{N} \{ &\text{cap}(x_n, p_i) \wedge \mathbb{1}[\hat{y}_i^{(\text{leaf})} = y_n] \\
&+ \text{cap}(x_n, p_{i+1}) \wedge \mathbb{1}[\hat{y}_{i+1}^{(\text{leaf})} = y_n] \\
&- \text{cap}(x_n, p_j) \wedge \mathbb{1}[\hat{y}_j^{(\text{leaf})} = y_n] \}.
\end{aligned}
\tag{24}
$$

For any leaf $j$ and its two child leaves $i, i+1$, we always have

$$
\sum_{n=1}^{N} \text{cap}(x_n, p_i) \wedge \mathbb{1}[\hat{y}_i^{(\text{leaf})} = y_n] \leq \sum_{n=1}^{N} \text{cap}(x_n, p_j) \wedge \mathbb{1}[\hat{y}_j^{(\text{leaf})} = y_n],
$$

$$
\sum_{n=1}^{N} \text{cap}(x_n, p_{i+1}) \wedge \mathbb{1}[\hat{y}_{i+1}^{(\text{leaf})} = y_n] \leq \sum_{n=1}^{N} \text{cap}(x_n, p_j) \wedge \mathbb{1}[\hat{y}_j^{(\text{leaf})} = y_n]
$$

which indicates that $a_i \leq \frac{1}{N} \sum_{n=1}^{N} \text{cap}(x_n, p_i) \wedge \mathbb{1}[\hat{y}_i^{(\text{leaf})} = y_n]$ and $a_i \leq \frac{1}{N} \sum_{n=1}^{N} \text{cap}(x_n, p_{i+1}) \wedge \mathbb{1}[\hat{y}_{i+1}^{(\text{leaf})} = y_n]$. Therefore,

$$
\lambda \leq \frac{1}{N} \sum_{n=1}^{N} \text{cap}(x_n, p_i) \wedge \mathbb{1}[\hat{y}_i^{(\text{leaf})} = y_n],
$$

$$
\lambda \leq \frac{1}{N} \sum_{n=1}^{N} \text{cap}(x_n, p_{i+1}) \wedge \mathbb{1}[\hat{y}_{i+1}^{(\text{leaf})} = y_n].
$$

□

### H.8 Proof of Proposition B.1

*Proof.* Recall that the objective is $R(d, \mathbf{x}, \mathbf{y}) = \ell(d, \mathbf{x}, \mathbf{y}) + \lambda H$, where the misclassification error $\ell(d, \mathbf{x}, \mathbf{y})$ is given by

$$\ell(d, \mathbf{x}, \mathbf{y}) = \frac{1}{N} \sum_{n=1}^{N} \sum_{k=1}^{K} \mathrm{cap}(x_n, p_k) \wedge \mathbb{1}[\hat{y}_k^{(\mathrm{leaf})} \neq y_n].$$

Any particular tree uses a specific leaf, and therefore a single class label, to classify all points within a set of equivalent points. Thus, for a set of equivalent points $u$, the tree $d$ correctly classifies either points that have the majority class label, or points that have the minority class label. It follows that $d$ misclassifies a number of points in $u$ at least as great as the number of points with the minority class label. To translate this into a lower bound on $\ell(d, \mathbf{x}, \mathbf{y})$, we first sum over all sets of equivalent points, and then for each such set, count differences between class labels and the minority class label of the set, instead of counting mistakes:

$$\ell(d, \mathbf{x}, \mathbf{y})$$
$$= \frac{1}{N} \sum_{u=1}^{U} \sum_{n=1}^{N} \sum_{k=1}^{K} \mathrm{cap}(x_n, p_k) \wedge \mathbb{1}[\hat{y}_k^{(\mathrm{leaf})} \neq y_n] \wedge \mathbb{1}[x_n \in e_u]$$
$$\geq \frac{1}{N} \sum_{u=1}^{U} \sum_{n=1}^{N} \sum_{k=1}^{K} \mathrm{cap}(x_n, p_k) \wedge \mathbb{1}[q_u = y_n] \wedge \mathbb{1}[x_n \in e_u].$$

Next, because every datum must be captured by a leaf in the tree $d$, $\sum_{k=1}^{K} \mathrm{cap}(x_n, p_k) = 1$.

$$\ell(d, \mathbf{x}, \mathbf{y}) \geq \frac{1}{N} \sum_{u=1}^{U} \sum_{n=1}^{N} \mathbb{1}[x_n \in e_u] \, \mathbb{1}[y_n = q_u] = \sum_{u=1}^{U} \theta(e_u),$$

where the final equality applies the definition of $\theta(e_u)$ in (9). Therefore, $R(d, \mathbf{x}, \mathbf{y}) = \ell(d, \mathbf{x}, \mathbf{y}) + \lambda K$ $\geq \sum_{u=1}^{U} \theta(e_u) + \lambda K$. □

## I  Ablation Experiments

We evaluate how much each of our bounds contributes to OSDT's performance and what effect the scheduling metric has on execution. Table 2 provides experimental statistics of total execution time, time to optimum, total number of trees evaluated, number of trees evaluated to optimum, and memory consumption on the recidivism data set. The first row is the full OSDT implementation, and the others are variants, each of which removes a specific bound. While all the optimizations reduce the search space, the lookahead and equivalent points bounds are, by far, the most significant, reducing time to optimum by at least two orders of magnitude and reducing memory consumption by more than one order of magnitude. In our experiment, although the scheduling policy has a smaller effect, it is still significant – curiosity is a factor of two faster than the objective function and consumes 25% of the memory consumed when using the objective function for scheduling. All other scheduling policies, *i.e.*, the lower bound and the entropy, are significantly worse.

## J  Regularized BinOCT

Since BinOCT always produces complete binary trees of given depth, we add a regularization term to the objective function of BinOCT. In this way, regularized BinOCT (RBinOCT) can generate the same trees as OSDT. Following the notation of [20], we provide the formulation of regularized BinOCT:

$$\min \sum_{l,c} e_{l,c} + \lambda \sum_{l} \alpha_l \text{ s.t.} \tag{25}$$

Per-bound performance improvement (ProPublica data set)

| Algorithm variant | Total time (s) | Slow-down | Time to optimum (s) |
|---|---|---|---|
| All bounds | 14.70 | — | 1.01 |
| No support bound | 17.11 | $1.16\times$ | 1.09 |
| No incremental accuracy bound | 30.16 | $2.05\times$ | 1.13 |
| No accuracy bound | 31.83 | $2.17\times$ | 1.23 |
| No lookahead bound | 31721 | $2157.89\times$ | 187.18 |
| No equivalent points bound | >12475 | $>848\times$ | — |

| Algorithm variant | Total #trees evaluated | #trees to optimum | Mem (GB) |
|---|---|---|---|
| All bounds | 232402 | 16001 | .08 |
| No support bound | 279763 | 18880 | .08 |
| No incremental accuracy bound | 546402 | 21686 | .08 |
| No accuracy bound | 475691 | 19676 | .09 |
| No lookahead bound | 284651888 | 3078274 | 10 |
| No equivalent points bound | >77000000 | — | >64 |

Table 2: Per-bound performance improvement, for the ProPublica data set ($\lambda = 0.005$, cold start, using curiosity). The columns report the total execution time, time to optimum, total number of trees evaluated, number of trees evaluated to optimum, and memory consumption. The first row shows our algorithm with all bounds; subsequent rows show variants that each remove a specific bound (one bound at a time, not cumulative). All rows except the last one represent a complete execution, *i.e.*, until the queue is empty. For the last row ('No equivalent points bound'), the algorithm was terminated after running out of the memory (about $\sim$64GB RAM).

$$\forall n \quad \sum_f f_{n,f} = 1$$

$$\forall r \quad \sum_l l_{r,l} = 1$$

$$\forall l \quad \sum_c p_{l,c} = 1$$

$$\forall n, f, b \in bin(f) \quad M \cdot f_{n,f} + \sum_{r \in lr(b)} \sum_{l \in ll(n)} l_{r,l} + \sum_{t \in tl(b)} M \cdot t_{n,t} - \sum_{t \in tl(b)} M \leq M$$

$$\forall n, f, b \in bin(f) \quad M' \cdot f_{n,f} + \sum_{r \in rr(b)} \sum_{l \in rl(n)} l_{r,l} - \sum_{t \in tl(b)} M' \cdot t_{n,t} \leq M'$$

$$\forall n, f \quad M'' \cdot f_{n,f} + \sum_{\max_{t(f)} < f(r)} \sum_{l \in ll(n)} l_{r,l} + \sum_{f(r) < \min_{t(f)}} \sum_{l \in rl(n)} l_{r,l} \leq M"$$

$$\forall\, l, c \quad \sum_{r : C_r = c} l_{r,l} - M''' \cdot p_l \leq e_{l,c}$$

$$\forall\, l \sum_r l_{r,l} \leq R \cdot \alpha_l, \tag{26}$$

where $1 \leq n \leq N$, $1 \leq f \leq F$, $1 \leq r \leq R$, $1 \leq l \leq L$, $1 \leq c \leq C$. Variables $\alpha_l$, $f_{n,f}$, $t_{n,t}$, $p_{l,c}$ are binary, and $e_{l,c}$ and $l_{r,l}$ are continuous (see Table 3). Compared with BinOCT, we add a penalization term $\lambda \sum_l \alpha_l$ to the objective function (25) and a new constraint (26), where $\lambda$ is the same as that of OSDT, and $\alpha_l = 1$ if leaf $l$ is not empty and $\alpha_l = 0$ if leaf $l$ is empty. All the rest of the constraints are the same as those of BinOCT. We use the same notation as in the original BinOCT formulation [20].

Figure 7 shows the trees generated by regularized BinOCT and OSDT when using the same regularization parameter $\lambda = 0.007$. Although the two algorithms produce the same optimal trees, regularized BinOCT is much slower than OSDT. In our experiment, it took only 3.390 seconds to run the OSDT algorithm to optimality, while the regularized BinOCT algorithm had not finished running after 1 hour.

In Figure 8, we provide execution traces of OSDT and RBinOCT. OSDT converges much faster than RBinOCT. For some datasets, *i.e.*, FICO and Monk1, the total execution time of RBinOCT was several times longer than that of OSDT.

| Notation | Type | Definition |
|---|---|---|
| $n$ | index | internal (non-leaf) node in the tree, $1 \leq n \leq N$ |
| $l$ | index | leaf of the tree, $1 \leq l \leq L$ |
| $r$ | index | row in the training data, $1 \leq r \leq R$ |
| $f$ | index | feature in the training data, $1 \leq f \leq F$ |
| $c$ | index | class in the training data, $1 \leq c \leq C$ |
| $bin(f)$ | set | feature $f$'s binary encoding ranges |
| $lr(b)$ | set | rows with values in $b$'s lower range, $b \in bin(f)$ |
| $ur(b)$ | set | rows with values in $b$'s upper range |
| $tl(b)$ | set | $t_{n,t}$ variables for $b$'s range |
| $ll(n)$ | set | node $n$'s leaves under the left branch |
| $rl(n)$ | set | node $n$'s leaves under the right branch |
| $K$ | constant | the tree's depth |
| $N = 2^K - 1$ | constant | the number of internal nodes (not leaves) |
| $L = 2^K$ | constant | the number of leaf nodes |
| $F$ | constant | the number of features |
| $C$ | constant | the number of classes |
| $R$ | constant | the number of training data rows |
| $T$ | constant | the total number of threshold values |
| $T_f$ | constant | the number of threshold values for feature $f$ |
| $T_{max}$ | constant | maximum of $T_f$ over all features $f$ |
| $V_r^f$ | constant | feature $f$'s value in training data row $r$ |
| $C_r$ | constant | class value in training data row $r$ |
| $\min_t(f)$ | constant | feature $f$'s minimum threshold value |
| $\max_t(f)$ | constant | feature $f$'s maximum threshold value |
| $M$ | constant | minimized big-M value |
| $f_{n,f}$ | binary | node $n$'s selected feature $f$ |
| $t_{n,t}$ | binary | node $n$'s selected threshold $t$ |
| $p_{l,c}$ | binary | leaf $l$'s selected prediction class $c$ |
| $\alpha_l$ | binary | $\alpha_l = 1$ if leaf $l$ is not empty |
| $e_{l,c}$ | continuous | error for rows with class $c$ in leaf $l$ |
| $l_{r,l}$ | continuous | row $r$ reaches leaf $l$ |
| $\lambda$ | continuous | the regularization parameter |

Table 3: Summary of the notations used in RBinOCT.

(a) RBinOCT
(accuracy: 66.223%)

(b) OSDT
(accuracy: 66.223%)

Figure 7: The decision trees generated by Regularized BinOCT and OSDT on the Monk1 dataset. The two trees are exactly the same, but regularized BinOCT is much slower in producing this tree.

# K    CART

We provide the trees generated by CART on COMPAS, Tic-Tac-Toe, and Monk1 datasets. With the same number of leaves as their OSDT counterparts (Figure 4, Figure 6, Figure 5), these CART trees perform much worse than the OSDT ones.

Figure 8: Execution traces of OSDT and regularized BinOCT. OSDT converges much more quickly than RBinOCT.

## L Cross-validation Experiments

The difference between training and test error is probabilistic and depends on the number of observations in both training and test sets, as well as the complexity of the model class. The best learning-theoretic bound on test error occurs when the training error is minimized for each model class, which in our case, is the maximum number of leaves in the tree (the level of sparsity). By adjusting the regularization parameter throughout its full

(a) COMPAS
(accuracy: 66.135%)

(b) MONK1
(accuracy: 91.935%)

(c) Tic-Tac-Toe
(accuracy: 76.513%)

Figure 9: Decision trees generated by CART on COMPAS, Tic-Tac-Toe, and Monk1 datasets. They are all inferior to these trees produced by OSDT as shown in Section 4.

range, OSDT will find the most accurate tree for each given sparsity level. Figures 10-16 show the training and test results for each of the datasets and for each fold. As indicated by theory, higher training accuacy for the same level of sparsity tends to yield higher test accuracy in general, but not always. There are some cases, like the car dataset, where OSDT's almost-uniformly-higher training accuracy leads to higher test accuracy, and other cases where all methods perform the same. In the case where all methods perform the same, OSDT provides a certificate of optimality showing that no better training performance is possible for the same level of sparsity.

Figure 10: 10-fold cross-validation experiment results of OSDT, CART, BinOCT on COMPAS dataset. Horizontal lines indicate the accuracy of the best OSDT tree in training.

Figure 11: 10-fold cross-validation experiment results of OSDT, CART, BinOCT on FICO dataset. Horizontal lines indicate the accuracy of the best OSDT tree in training.

Figure 12: 10-fold cross-validation experiment results of OSDT, CART, BinOCT on Tic-Tac-Toe dataset. Horizontal lines indicate the accuracy of the best OSDT tree in training.

Figure 13: 10-fold cross-validation experiment results of OSDT, CART, BinOCT on car dataset. Horizontal lines indicate the accuracy of the best OSDT tree in training.

Figure 14: 10-fold cross-validation experiment results of OSDT, CART, BinOCT on Monk1 dataset. Horizontal lines indicate the accuracy of the best OSDT tree in training.

Figure 15: 10-fold cross-validation experiment results of OSDT, CART, BinOCT on Monk2 dataset. Horizontal lines indicate the accuracy of the best OSDT tree in training.

Figure 16: 10-fold cross-validation experiment results of OSDT, CART, BinOCT on Monk3 dataset. Horizontal lines indicate the accuracy of the best OSDT tree in training.



[Supplementary Material 2 · OSDT_NIPS_Poster.pdf]

# Optimal Sparse Decision Trees

Xiyang Hu[1], Cynthia Rudin[2], Margo Seltzer[3*]

xiyanghu@cmu.edu, cynthia@cs.duke.edu, mseltzer@cs.ubc.ca
[1]Carnegie Mellon University, [2]Duke University, [3]The University of British Columbia
*Author names are in alphabetic order.

## Overview

- Decision Trees: Extremely popular form for interpretable ML models since the 1980's.
- Existing algorithms use greedy splitting and pruning, providing no guarantee of optimality.
- OSDT is the first practical algorithm for construction of optimal decision trees for binary variables.
- OSDT combines analytical bounds, computational caching, and fast bit-vector operations to efficiently prune the search space.

## Notation

We focus on binary classification, and our decision trees are Boolean functions.

- A tree can be expressed in terms of its leaves.
- A leaf, $p_k$, is the classification rule of the path from the root to leaf $k$.
- Let $H$ be the number of leaves in a tree and $K <= H$ be the number of leaves that will not be split.
- We represent a decision tree, $d$ as $(d_{un}, \delta_{un}, d_{split}, \delta_{split}, K, H)$, where
  - $d_{un} = (p_1, \ldots, p_K)$ are the unchanged leaves of $d$,
  - $\delta_{un} = (\hat{y}_1^{(leaf)}, \ldots, \hat{y}_K^{(leaf)}) \in \{0,1\}^K$ are the predicted labels of leaves $d_{un}$,
  - $d_{split} = (p_{K+1}, \ldots, p_H)$ are the leaves we are going to split, and
  - $\delta_{split} = (\hat{y}_{K+1}^{(leaf)}, \ldots, \hat{y}_H^{(leaf)}) \in \{0,1\}^{H-K}$ are the predicted labels of leaves $d_{split}$.

## Objective Function

For a tree $d = (d_{un}, \delta_{un}, d_{split}, \delta_{split}, K, H)$, we define its objective function as a combination of the misclassification error and a sparsity penalty on the number of leaves:

$$R(d, \mathbf{x}, \mathbf{y}) = \ell(d, \mathbf{x}, \mathbf{y}) + \lambda H(d). \qquad (1)$$

where $R(d, \mathbf{x}, \mathbf{y})$ is a regularized empirical risk, $H(d)$ is the number of leaves in the tree $d$, and the loss $\ell(d, \mathbf{x}, \mathbf{y})$ is the misclassification error of $d$, i.e., the fraction of training data with incorrectly predicted labels.

## Optimization Framework

We minimize the objective function based on a branch-and-bound framework. We prove a series of useful bounds that work together to eliminate a large part of the search space.

### Hierarchical objective lower bound

Tree T
lower bound b(T)

any child tree t of Tree T
Objective R(t)≥b(T)

## Optimization Framework Cont'd

### Objective lower bound with one-step lookahead

Tree T
If b(T)+λ≥R$^c$

any child tree t of Tree T
Objective R(t)≥b(T)+λ≥R$^c$

### Lower bound on node support

In an optimal tree, for any internal node n, its support supp(n)≥2λ.

### Lower bound on incremental classification accuracy

Tree T
split one leaf

get two new leaves
Incremental accuracy≥λ

### Leaf accurate support bound

In an optimal tree, for any leaf l, its classification accuracy≥λ.

### Leaf permutation bound

Tree T$_1$          Tree T$_2$

T$_1$ and T$_2$ are actually the same, only need to evaluate one of them.

### Equivalent points bound

For a given dataset, if there are multiple samples with exactly the same features but different labels, then no matter how we build our classifier, we will always predict some of these points incorrectly.

## Algorithm

The loss can be decomposed into two parts corresponding to the unchanged leaves and the leaves to be split:

- $\ell(d, \mathbf{x}, \mathbf{y}) \equiv \ell_p(d_{un}, \delta_{un}, \mathbf{x}, \mathbf{y}) + \ell_p(d_{split}, \delta_{split}, \mathbf{x}, \mathbf{y})$, where $d_{un} = (p_1, \ldots, p_K)$, $\delta_{un} = (\hat{y}_1^{(leaf)}, \ldots, \hat{y}_K^{(leaf)})$, $d_{split} = (p_{K+1}, \ldots, p_H)$ and $\delta_{split} = (\hat{y}_{K+1}^{(leaf)}, \ldots, \hat{y}_H^{(leaf)})$;
- $\ell_p(d_{un}, \delta_{un}, \mathbf{x}, \mathbf{y}) = \frac{1}{N}\sum_{n=1}^N \sum_{k=1}^K \text{cap}(x_n, p_k) \wedge \mathbb{1}[\hat{y}_k^{(leaf)} \neq y_n]$ is the proportion of data in the unchanged leaves that are misclassified;
- $\ell_p(d_{split}, \delta_{split}, \mathbf{x}, \mathbf{y}) = \frac{1}{N}\sum_{n=1}^N \sum_{k=K+1}^H \text{cap}(x_n, p_k) \wedge \mathbb{1}[\hat{y}_k^{(leaf)} \neq y_n]$ is the proportion of data in the leaves we are going to split that are misclassified;
- Define a lower bound $b(d_{un}, \mathbf{x}, \mathbf{y})$ on the objective by leaving out the latter loss,

$$b(d_{un}, \mathbf{x}, \mathbf{y}) \equiv \ell_p(d_{un}, \delta_{un}, \mathbf{x}, \mathbf{y}) + \lambda H \leq R(d, \mathbf{x}, \mathbf{y}), \qquad (2)$$

where the leaves $d_{un}$ are kept and the leaves $d_{split}$ are going to be split. Here, $b(d_{un}, \mathbf{x}, \mathbf{y})$ gives a lower bound on the objective of *any* child tree of $d$.

---

**Algorithm 1** Branch-and-bound for learning optimal decision trees.

**Input:** Objective function $R(d, \mathbf{x}, \mathbf{y})$, objective lower bound $b(d_{un}, \mathbf{x}, \mathbf{y})$, set of features $S = \{s_m\}_{m=1}^M$, training data $(\mathbf{x}, \mathbf{y}) = \{(x_n, y_n)\}_{n=1}^N$, initial best known tree $d^0$ with objective $R^0 = R(d^0, \mathbf{x}, \mathbf{y})$; $d^0$ could be obtained as output from another (approximate) algorithm, otherwise, $(d^0, R^0) = (\text{null}, 1)$ provides reasonable default values. The initial value of $\delta_{split}$ is the majority label of the whole dataset.
**Output:** Provably optimal decision tree $d^*$ with minimum objective $R^*$

$(d^c, R^c) \leftarrow (d^0, R^0)$    ▷ Initialize best tree and objective
$Q \leftarrow \text{queue}([\, (\langle\rangle, \langle\rangle, \langle\rangle, \delta_{split}, 0, 0)\, ])$    ▷ Initialize queue with empty tree
**while** $Q$ not empty **do**    ▷ Stop when queue is empty
  $d = (d_{un}, \delta_{un}, d_{split}, \delta_{split}, K, H) \leftarrow Q.\text{pop}()$    ▷ Remove tree $d$ from the queue
  **if** $b(d_{un}, \mathbf{x}, \mathbf{y}) < R^c$ **then**    ▷ **Bound:** Hierarchical objective lower bound
    $R \leftarrow R(d, \mathbf{x}, \mathbf{y})$    ▷ Compute objective of tree $d$
    **if** $R < R^c$ **then**    ▷ Update best tree and objective
      $(d^c, R^c) \leftarrow (d, R)$
    **end if**
    **for** every possible combination of features to split $d_{split}$ **do**
       ▷ **Branch:** Enqueue $d_{un}$'s children
      split $d_{split}$ and get new leaves $d_{new}$
      **for** each possible subset $d'_{split}$ of $d_{new}$ **do**
        $d'_{un} = d_{un} \cup (d_{new} \setminus d'_{split})$
        $Q.\text{push}((d'_{un}, \delta'_{un}, d'_{split}, \delta'_{split}, K', H'))$
      **end for**
    **end for**
  **end if**
**end while**
$(d^*, R^*) \leftarrow (d^c, R^c)$    ▷ Identify provably optimal solution

---

## Incremental Computation

During the execution of our algorithm, for each tree $d$, we compute the lower bound $b(d_{un}, \mathbf{x}, \mathbf{y})$ of the tree based on its unchanged leaves $d_{un}$ and the corresponding objective $R(d, \mathbf{x}, \mathbf{y})$ of the tree. Given the hierarchical nature of the parent-children relationship, we *incrementally* compute the objective function and the lower bound throughout the brand-and-bound execution of the algorithm. Together, these ideas save >97% execution time.

## Experiments

- Accuracy and optimality

Training accuracy of OSDT, CART, BinOCT on different data (time limit: 30min). Horizontal lines indicate the accuracy of the best OSDT tree. On most datasets, all trees of BinOCT and CART are below this line.

- Convergence

Example OSDT execution traces (COMPAS Dataset, $\lambda = 0.005$). Lines are the objective value and dashes are the lower bound for OSDT. For each scheduling policy, we mark the time to optimum and the optimal objective value using a star.

- Scalability

Scalability with respect to number of samples and number of features using (multiples of) the ProPublica data set. ($\lambda = 0.005$).

## Sample Trees

Figure: The optimal decision tree generated by OSDT on COMPAS dataset. ($\lambda = 0.005$)

(a) BinOCT (accuracy: 76.722%)    (b) OSDT (accuracy: 82.881%)

Figure: The decision tree generated by BinOCT and OSDT on the Tic-Tac-Toe data. Trees of BinOCT must be complete binary trees, while OSDT can generate trees of any shape.

(a) BinOCT (accuracy: 91.129%)    (b) OSDT (accuracy: 100%)

Figure: The decision tree generated by BinOCT and OSDT on Monk1 dataset. The tree generated by BinOCT includes useless splits, while OSDT can avoid this problem.

## Paper and Code

- Paper: https://arxiv.org/abs/1904.12847
- Code: https://github.com/xiyanghu/OSDT

[Supplementary Material 3]

# Optimal Sparse Decision Trees
## Xiyang Hu, Cynthia Rudin, Margo Seltzer

# Optimal Sparse Decision Trees

# Optimal Sparse Decision Trees

# Optimal Sparse Decision Trees

# Optimal Sparse Decision Trees

Data → **Mathematical Programming Solvers** →

rain?
- Y → construction?
  - Y → traffic
  - N → no traffic
- N → rush hour?
  - Y → Construction?
    - Y → traffic
    - N → Friday?
      - Y → no traffic
      - N → traffic
  - N → no traffic

# Optimal Sparse Decision Trees

# Optimal Sparse Decision Trees

# Optimal Sparse Decision Trees

Data →

Neural Networks (crossed out)

rain?
- Y → construction?
  - Y → traffic
  - N → no traffic
- N → rush hour?
  - Y → Construction?
    - Y → traffic
    - N → Friday?
      - Y → no traffic
      - N → traffic
  - N → no traffic

# Optimal Sparse Decision Trees

$$\min_{\text{tree}} \hat{L}(\text{tree}, \{(x_i, y_i)\}_i) \text{ where}$$

$$\hat{L}(\text{tree}, \{(x_i, y_i)\}_i) = \frac{1}{n} \sum_{i=1}^{n} 1_{[\text{tree}(x_i) \neq y_i]} + C(\# \text{ leaves in tree})$$

<span style="color:crimson">Misclassification error</span>　　<span style="color:crimson">Sparsity</span>

We solve this to optimality.
No greedy splitting and pruning like C4.5 and CART
The key: very efficient branch & bound combined with computer systems.

# Optimal Sparse Decision Trees

$$\min_{\text{tree}} \hat{L}(\text{tree}, \{(x_i, y_i)\}_i) \text{ where}$$

$$\hat{L}(\text{tree}, \{(x_i, y_i)\}_i) = \frac{1}{n} \sum_{i=1}^{n} 1_{[\text{tree}(x_i) \neq y_i]} + C(\# \text{ leaves in tree})$$

Misclassification error          Sparsity

Prior offenses > 3
- no / yes
- yes → Predict Arrest

Age< 26
- no
- yes

Prior offenses > 1
- no
- yes → Predict Arrest

no → Predict No Arrest

Any juvenile crimes
- no → Predict No Arrest
- yes → Predict Arrest

← An example of an optimal tree on the re-arrest data

# Optimal Sparse Decision Trees

Analytical Bounds Reduce the Search Space

This collection of  theorems show that some partial trees can never be extended to form optimal trees.

Not enough data

Not accurate data

Too many leaves

# Optimal Sparse Decision Trees

## Represent a tree by its leaves

rain & construction & traffic

rain & no construction & no traffic

no rain & rush hour & construction & traffic

no rain & rush hour & no construction & Friday and no traffic

no rain & rush hour & no construction & Friday and traffic

no rain & no rush hour & no traffic

# Optimal Sparse Decision Trees

## Permutation map: Discover identical trees already evaluated

rain & construction & traffic

rain & no construction & no traffic

no rain & rush hour & construction & traffic

no rain & rush hour & no construction & Friday and no traffic

no rain & rush hour & no construction & Friday and traffic

no rain & no rush hour & no traffic

# Optimal Sparse Decision Trees

Bit-vectors describe data represented by each leaf

rain & construction & traffic
[1000010001001110000..........................0]
rain & no construction & no traffic
[0110001000000000110.........................1]
no rain & rush hour & construction & traffic
[0001000100000001000.........................0]
no rain & rush hour & no construction & Friday and no traffic
[0000100000000000001.........................0]
no rain & rush hour & no construction & Friday and traffic
[0000000010000000000.........................0]
no rain & no rush hour & no traffic
[0000000000011000000.........................0]

rain?
Y    N

construction?
Y    N

rush hour?
Y    N

traffic    no traffic    construction?    no traffic

Y    N

traffic    Friday?

Y    N

no traffic    traffic

# Optimal Sparse Decision Trees

## Incremental computation of objective and bounds

# Optimal Sparse Decision Trees

Strong analytical bounds

**+**

Leaf-based representation

**+**

Permutation map

**+**     **=**     Fast Implementation

Caching of intermediate results

**+**

Incremental computation

**Classification Accuracy of Monk1 Dataset**

**Classification Accuracy of Monk1 Dataset**

**Classification Accuracy of Monk2 Dataset**

**Classification Accuracy of Monk3 Dataset**

**Classification Accuracy of COMPAS Dataset**

**Classification Accuracy of Tic-Tac-Toe Dataset**

**Classification Accuracy of Car Dataset**

# Monk 1 dataset

(a) BinOCT (accuracy: 91.129%)

(b) OSDT (accuracy: 100%)

# COMPAS dataset

```
                    priors>3
                   /        \
              age<26          Yes
             /      \
           No        priors:2-3
                     /          \
        juvenile-crimes=0        Yes
              /    \
           Yes      No
```

**Execution Traces of OSDT, RBinOCT and CART (tictactoe Dataset)**

Legend:
- LowerBound-OSDT
- Objective-OSDT
- Objective-RBinOCT
- LowerBound-RBinOCT
- Objective-CART

# Summarize

- First practical method for optimal sparse binary split decision trees
- Current work:
  - Non-straightforward speedup for continuous variables
  - Generalization to other objectives

Xiyang Hu, Cynthia Rudin, Margo Seltzer. Optimal Sparse Decision Trees.
https://arxiv.org/abs/1904.12847