[Reviews · NeurIPS 2019]

Reviewer 1



-Figure 1 of the evaluation section only has OSDT-WarmStart; why is OSDT missing? -A runtime benchmark against BinOCT and CART needs to be included in the evaluation section as well. -Figure 4 shows the optimal tree generated by OSDT on COMPAS; what do BinOCT and CART generate? Other nits: Line 98: the upper bound of the sum should be "D - 1", instead of "d - 1". Line 124: "alternately" should be "alternatively"

Reviewer 2



Originality: This work is an improvements on existing methods. The originality comes from identifying various bounds that one can impose to prune the search space. It is clear how this work differs from existing works. Quality: The work is technically correct. The experiments demonstrate the claims that the method is better at finding optimal trees than existing methods. The experiments seem thorough though there is no description of the weaknesses of the method. Clarity: The paper has been written in a manner that is straightforward to read and follow. Significance: There are two factors which dent the significance of this work. 1. The work uses only binary features. Real world data is usually a mix of binary, real and categorical features. It is not clear if the method is applicable to real and categorical features too. 2. The method does not seem to be scalable, unless a distributed version of it is developed. It's not reasonable to expect a single instance can hold all the training data that the real world datasets ususally contain.

Reviewer 3



Originality: Training of optimal decision trees is clearly a problem that has seen a lot of prior work. A distinguishing feature of this submission is that it focuses on optimal *sparse* decision trees for binary variables, and that the approach seems to be feasible in practice, which is achieved by a combination of analytical bounds that reduce the search space as well as efficient implementation techniques. The work builds upon the CORLES algorithm and its approach to creating optimal decision lists. However, the authors extend this approach to decision trees in a non-trivial manner that adds substantial novelty. Quality: The claims of the paper are very well supported by theoretical analysis as well as experiments. Besides a quantitative comparison, the experiments also give some insight into what optimal decision trees look like, and where non-optimal approaches fail. Clarity: The manuscript is very well-written and organized. The main manuscript is very condensed, which can make it hard to read at times; but the supplementary includes all relevant material at great detail. Significance: The paper mainly focuses on improved training accuracy. To further underline the practical relevance of this algorithm, the authors would need to demonstrate convincingly that improved training accuracy also translates into improved test set generalization. This can be harder to demonstrate than training accuracy, because it depends heavily on the data, the number of leaves, etc. The paper has no claims in this direction, instead focusing on the training problem. The supplementary material does provides numbers for accuracy on the test set for a number of problems, but the results are less conclusive than for training.

[Author Response · NeurIPS 2019]

Thank you reviewers for your positive and helpful suggestions!

**R1:** Figure 1 no OSDT (warm-start only): They provably have the same solution; the two points would be identical. Runtime benchmark against BinOCT and CART: Good idea. We can add BinOCT and CART execution profiles to figures like that shown in the paper's Figure 2. (We include examples in Figure 1 below.) In general, OSDT converges and *completes more quickly* than BinOCT; BinOCT *sometimes includes redundant leaves* leading to non-optimal solutions to our problem. CART frequently completes very quickly, but does not always yield results near optimal. For example, on monk-1, monk-2 and tic-tac-toe, the results of CART are far away from the optimal ones. Show BinOCT and CART trees as well as OSDT trees (Fig 4): Sure! Figures 5 and 6 in the supplementary materials have example binOCT trees; we can add CART trees as well. Revised title: We've made it efficient for real-valued features since the submission, owing to a good implementation of Theorem F.1. (see Figure 2 below). Introduce notation gradually: easy, will do. Including more background on CORELS: easy, will do. Thank you for all your suggestions! We appreciate it!

**R2:** Paper contributions: Sorry! (1) first practical optimal decision tree algorithm to achieve provably optimal solutions for nontrivial problems. (2) a series of new analytical bounds to reduce the space (Sec 3.2). (3) the first practical algorithmic use of a tree representation using only its leaves (Sec 3). (4) Implementation speedups saving 97% of run time (Sec 4 and appendix). (5) For the important COMPAS and FICO datasets (Sec 5), optimal trees have never previously been published. We present the first ones.
Utility for real and categorical features too: The COMPAS and FICO datasets used in the paper both have real-valued features, so it is applicable. Since the submission, our implementation of Thm F.1 is more efficient, allowing better scaling with real-valued features (e.g. on the Iris dataset in Figure 2, we reduced the number of tree evaluations by 45%). All possible split points are considered for each real-valued variable in our current implementation.

Scalability: We have been working on a parallel implementation and expect to have scalability results from it in time for the camera ready submission. Parallelization is *much* less difficult than what we already did.

Figure 2: OSDT's result on Iris dataset. Covariates are real-valued. OSDT considered all possible splits to produce the optimal decision boundary (black).

**R3:** The authors would need to demonstrate convincingly that improved training accuracy also translates into improved test set generalization: The basis for all of statistical learning theory is that training accuracy *and simplicity* provably lead to better test accuracy, and OSDT's objective incorporates both accuracy and sparsity. The 10-fold cross validation results in the supplementary materials (Section J) illustrate this, showing in-sample and out-of-sample accuracy.
The results are less conclusive than for training: Since OSDT found solutions that were as accurate as other methods in testing, but were more sparse, we interpret that as conclusive evidence. We will discuss this in the main body.

Figure 1: Execution traces of OSDT, CART and regularized BinOCT. OSDT converges much faster than RBinOCT generally. Traces end when execution completes for each algorithm separately. Runs were stopped after 10 min.

[Meta-Review · NeurIPS 2019]

Reviewers are very positive about the paper. The contribution is clear and significant. The paper should clearly be accepted. The authors should take into account all reviewers' comments when preparing the final version of their paper, as promised in their response, in particular the improvements suggested by reviewer 1 (as I agree that the paper is heavy on notation and not totally self-contained).